# Low-Rank Quantization-Aware Training for LLMs

## Abstract

Quantization-aware training (QAT) methods, generally produce the best model performance, however it comes at the cost of excessive memory usage and runtime, making it impractical when applying for LLMs. Inspired by parameter-efficient fine-tuning (PEFT) literature, we propose **L**ow-**R**ank **QAT** – a lightweight and memory-efficient QAT algorithm for LLMs. LR-QAT employs several components to save memory without sacrificing predictive performance: (a) low-rank quantization-aware reparameterization; (b) downcasting operation using fixed-point or double-packing and (c) checkpointing. Unlike most related work, our method (i) is inference-efficient, leading to no additional overhead compared to traditional post-training quantization (PTQ); (ii) can be seen as a general extended pretraining framework, meaning that the resulting model can still be utilized for any down-stream task afterwards; (iii) is orthogonal to most of recent PTQ methods and thus can be seamlessly combined with them. We apply LR-QAT to LLaMA-1/2/3 and Mistral model families and validate its effectiveness on several downstream tasks. Our method outperforms most of recent LLM quantization approaches and reaches the same model performance as full-model QAT at the fraction of its memory usage. Specifically, we can train a 7B LLM on a single consumer grade GPU with 24GB of memory.

## 1 Introduction

In recent years, large language models (LLMs) have emerged as a powerful tool for a plethora of natural language processing tasks. As these models continue to grow in size and capability, addressing their ever increasing computational and memory demands becomes crucial for practical deployment, especially when considering resource-constrained edge devices.

One of the most effective methods to tackle this problem is neural network quantization, which uses low-bit precision for weight and activation tensors. While recent post-training quantization (PTQ) methods can help with decreasing the model size and improving the computational efficiency of LLMs, they typically lead to subpar performance, especially in the case of low-bit ($\leq 4$) quantization. Quantization-aware training (QAT), conversely, yields significantly better model performance compared to PTQ. However, due to extreme model sizes of modern LLMs, using traditional QAT is very computationally expensive and requires a prohibitively high GPU memory usage, making it impractical.

Inspired by parameter-efficient fine-tuning (PEFT) and low-rank adaptation (LoRA) literature, we propose **Low-Rank Quantization-Aware Training** (**LR-QAT**) – a lightweight memory-efficient and inference-efficient QAT algorithm for LLMs. LR-QAT reduces the memory requirements of training a 7B LLM from >98GB of GPU memory to <21GB without degrading the predictive performance compared to traditional full-model QAT, making it possible to train such models on a single consumer grade GPU. Unlike most related work that combines low-rank adaptation with quantization, our method is also *inference-efficient*. After the training is complete, the auxiliary matrices are naturally absorbed into the quantized weight tensor without loss of accuracy and no extra overhead at inference time. LR-QAT is positioned as a general *extended pretraining* method, as opposed to being strictly a fine-tuning method – the resulting model is a low-bit general pretrained LLM, that can still be utilized for any task afterwards. If needed, our resulting low-bit pretrained LLM can be fine-tuned on specific downstream tasks or used with multiple LoRA adapters for rapid switching between various tasks.

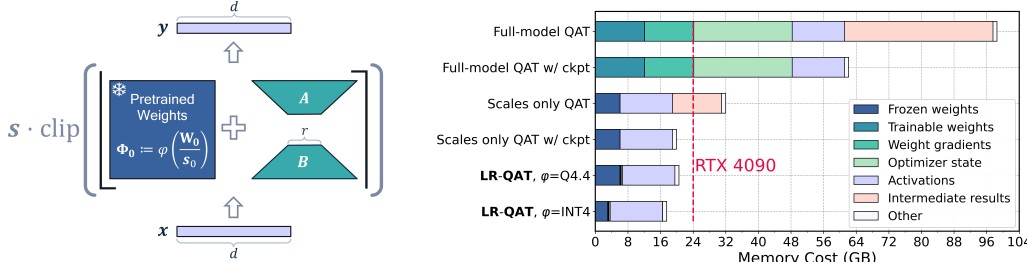

Figure 1: *Left:* A schematic illustration of our proposed LR-QAT. $\mathbf{x}$ and $\mathbf{y}$ denote the input and the output of the linear layer. *Right:* Memory requirements for training with various QAT techniques on LLaMA-2 7B, assuming batch size 1, sequence length 1024, rank $r = 32$, and BF16 compute data type. 'Intermediate results' refer to the results of some intermediate computations, *e.g.* after rounding/clipping in (3), which are saved in memory for the backward pass.

LR-QAT introduces and combines several innovations designed to reduce memory use without sacrificing model performance: (1) a form of **QAT with low-rank reparameterization**, in which we place the low-rank weights in the integer domain to ensure they align with the quantization grid of the pretrained weights. This allows for seamless fusion during inference into a single low-bit integer matrix. (2) A **downcasting operator** that represents the frozen pretrained weights as low-bit INT-$b$ ($b \leq 4$) double-packed into INT8 or as fixed-point values stored in INT8. (3) Finally, we combine the proposed quantization formulation with **gradient checkpointing** to avoid aggressive memory spikes from storing some of the intermediate results in memory for the backward pass.

We apply LR-QAT to LLaMA-1/2/3 and Mistral model families and demonstrate its effectiveness on several general language modeling datasets and zero-shot evaluation on some of the common reasoning downstream tasks. Our method outperforms recent LLM quantization approaches and reaches the same predictive performance as full-model QAT at the fraction of its memory usage. Finally, our method can be applied across a wide range of quantization settings, including per-channel or per-block weight quantization, activation quantization, and can be combined with most of other PTQ techniques.

## 2 BACKGROUND AND RELATED WORK

Neural network quantization is one of the most powerful ways to reduce model footprint, data transfer and compute requirements. By quantizing a model, high bit-width floating point weights and activations can be represented using low-bit numbers. On top of that, by using low-bit fixed-point representations, such as INT8, one can further reduce energy consumption since the fixed-point operations are more efficient than their floating-point counterparts. Quantizing to 8 bits or lower, however, typically introduces quantization noise in the model, resulting in a potential drop in accuracy/perplexity.

In this section, we provide a brief overview of uniform affine quantization and a summary of recent methods for LLM quantization. We will discuss some of the trade-offs of those techniques. Finally, we touch upon the challenges of LLM quantization and some of the limitations of current approaches.

**Uniform affine quantization**    We use the following definition of the quantization function:

$$\widehat{\mathbf{x}} := q\left(\mathbf{x}; \mathbf{s}, \mathbf{z}, b\right) = \mathbf{s} \cdot \Big( \underbrace{\text{clip}\Big(\Big\lfloor \frac{\mathbf{x}}{\mathbf{s}} \Big\rceil + \mathbf{z}; -2^{b-1}, 2^{b-1} - 1 \Big)}_{=: \, \mathbf{x}_{\mathbb{Z}}} - \mathbf{z} \Big), \tag{1}$$

where $\mathbf{x}$ denotes the quantizer input (i.e., network weights or activations), $\mathbf{s}$ the higher precision quantization scale, $\mathbf{z}$ the integer zero offset, and $b$ the bitwidth. $\lfloor \cdot \rceil$ denotes the round-to-nearest-integer operator. Quantization parameters $\mathbf{s}$, $\mathbf{z}$ can be shared across the components of $\mathbf{x}$ (typically per-channel or block-wise). One can see that such a quantizer approximates an original floating point vector as $\mathbf{x} \approx \mathbf{s} \cdot (\mathbf{x}_{\mathbb{Z}} - \mathbf{z})$, where each element in $\mathbf{x}_{\mathbb{Z}}$ is a $b$-bit integer value. This quantization scheme is called *uniform affine* or *asymmetric* quantization (Hubara et al., 2017; Krishnamoorthi, 2018; Zhou et al., 2016) and is one of the most commonly used quantization schemes because it

allows for efficient implementation of fixed-point arithmetic. In the case of *symmetric* quantization, we restrict the quantization grid to be symmetric around $\mathbf{z} = \mathbf{0}$.

**Post-training quantization methods**   Post-training quantization (PTQ) algorithms take a pretrained high precision (FP32 / FP16 / BF16) network and convert it directly into a fixed-point network without the need for the original training pipeline (Banner et al., 2018; Cai et al., 2020; Choukroun et al., 2019; Hubara et al., 2020; Krishnamoorthi, 2018; Li et al., 2021; Meller et al., 2019; Nagel et al., 2019; 2020; Zhao et al., 2019). These methods are either data-free or only require a small calibration dataset and are generally quite easy to use. Having almost no hyperparameter tuning makes them usable via a single API call as a black-box method to quantize a pretrained neural network in a computationally efficient manner.

Post-training quantization of LLMs is a challenging task due to presence of numerical outliers in weights and activations (Bondarenko et al., 2021; 2024; Kovaleva et al., 2021; Dettmers et al., 2022; Sun et al., 2024). Existing LLM PTQ methods can be broadly categorized into *weights-only* quantization and *weight-activation* quantization.

Weights-only quantization focuses on converting weights to low-bit values. For instance, GPTQ (Frantar et al., 2022) employs second-order information to iteratively round grouped weights and correct the quantization error in the remaining groups. SpQR (Dettmers et al., 2023), AWQ (Lin et al., 2023) and OWQ (Lee et al., 2024) emphasize the importance of so-called "salient" weights that correspond to high-magnitude activations. Other recent W-only methods include (Jeon et al., 2023; Lee et al., 2023b; Luo et al., 2023; Chee et al., 2024).

Weight-activation quantization compresses both weights and activations. SmoothQuant (Xiao et al., 2023), `LLM.int8()` (Dettmers et al., 2022) and Outlier Suppression (Wei et al., 2022) achieve W8A8 quantization by managing activation outliers. `LLM.int8()` uses mixed-precision decomposition, while the other two employ channel-wise scaling. OmniQuant (Shao et al., 2023) modulates the extreme values of weights by optimizing the clipping threshold and shifts the challenge of quantization from activations to weights by employing the learnable equivalent transformation. Some of the other recent W&A PTQ methods are (Lee et al., 2023a; Liu et al., 2023a; Wei et al., 2023; Yuan et al., 2023; Tang et al., 2024; Yao et al., 2022; Lin et al., 2024).

**Quantization-aware training methods**   Quantization-aware training (QAT) methods (Bhalgat et al., 2020; Esser et al., 2020; Gupta et al., 2015; Jacob et al., 2018; Krishnamoorthi, 2018) simulate quantization during training, allowing the model to find more optimal solutions compared to PTQ approaches. However, better accuracy/perplexity comes at the cost of neural network training, i.e., longer training times, increased memory usage, need for labeled data and hyperparameter search.

The excessive training cost and memory usage of traditional QAT methods make them unsuitable for quantizing modern LLMs. A few works that apply QAT to LLMs include LLM-QAT (Liu et al., 2023b) and BitDistiller (Du et al., 2024), both of which explore the application of knowledge distillation within QAT setting. Additionally, EdgeQAT (Shen et al., 2024) investigates the application of QAT to tiny language models (those with fewer than 100 million parameters).

**Low-rank adapters for fine-tuning**   Low-rank adaptation (LoRA) (Hu et al., 2021) is a parameter efficient fine-tuning (PEFT) method that reduces memory requirements. LoRA freezes the pretrained weights $\boldsymbol{W} = \boldsymbol{W_0}$, and only trains a small set of low-rank trainable parameters, often termed *adapters*. Given a linear projection $\mathbf{y} = \boldsymbol{W_0}\mathbf{x}$ with $\boldsymbol{W_0} \in \mathbb{R}^{m \times k}$, LoRA computes

$$\mathbf{y} = \boldsymbol{W_0}\mathbf{x} + \frac{\alpha}{r}\boldsymbol{A}\boldsymbol{B}\mathbf{x}, \tag{2}$$

where $\boldsymbol{A} \in \mathbb{R}^{m \times r}$, $\boldsymbol{B} \in \mathbb{R}^{r \times k}$, $r < \min\{m, k\}$ – rank, and $\alpha$ is a scalar that is constant in $r$. The benefits of LoRA are that it is much cheaper and often performs on par with or better than full fine-tuning. Additionally, the fine-tuned (floating-point) model can be deployed without extra cost, as the low-rank matrices can be fused into the pretrained weights after fine-tuning ($\boldsymbol{W} := \boldsymbol{W_0} + \frac{\alpha}{r}\boldsymbol{A}\boldsymbol{B}$).

Naturally, there have been several works that explored the combination of LoRA and quantization. QLoRA (Dettmers et al., 2024) quantizes the pretrained weights to 4 bit using (a non-uniform) NF4 format and dequantizes them in the forward pass to further reduce fine-tuning memory footprint. LoftQ (Li et al., 2023) proposed an iterative SVD-based procedure for initializing $\boldsymbol{A}$, $\boldsymbol{B}$ that yields

| Method | Accuracy | Memory efficiency | Inference efficiency |
|---|:---:|:---:|:---:|
| PTQ | ✗ | ✓ | ✓ |
| Full-model QAT | ✓ | ✗ | ✓ |
| QLoRA / LoRA-based | ✓ | ✓ | ✗ |
| **LR-QAT (ours)** | ✓ | ✓ | ✓ |

Table 1: A comparison between existing approaches and the proposed method.

faster fine-tuning convergence when used together with low-bit quantization. LQ-LoRA (Guo et al., 2023) further extends initialization technique from LoftQ to mixed precision and data aware cases. Other recent works include (Jeon et al., 2024; Zhang et al., 2024). QA-LoRA (Xu et al., 2023) uses INT4 quantization and is the only work we are aware of that attempts to fuse auxiliary LoRA weights back into the frozen $\boldsymbol{W}_{\mathbb{Z}}$. However, their method is designed to work with group-wise quantization with a small group size of 32 and hence cannot be applied to bigger group sizes or per-channel quantization, which are common settings for weights.

Finally, we consider the work closest to ours to be PEQA (Kim et al., 2024), that attempts to combine the benefits of inference-efficiency of QAT together with memory-efficiency of PEFT methods. Similar to our method and unlike QA-LoRA, it does not impose any restrictions on quantization granularity. However, just like most of LoRA-based methods, their approach focuses on a task-specific fine-tuning as opposed to being a general extended pretraining method. In addition, PEQA has significantly fewer degrees of freedom compared to our method, leading to subpar performance.

## 3 MOTIVATION

While generally fast and simple, PTQ suffers from limited performance in low-bit scenarios. Although QAT methods still perform well in low-bit regimes, their high training costs and memory usage make them impractical for LLMs (see Figure 1, right). Existing LoRA-based methods aim to address memory efficiency during fine-tuning. However, in most cases they do not consider efficient inference. Techniques such as QLoRA (Dettmers et al., 2024), which do not explicitly fuse the low-rank adapters $\boldsymbol{A}$ and $\boldsymbol{B}$, can incur up to 30% additional inference latency compared to the base model (Bhardwaj et al., 2024). This is in line with our own simulation for LLaMA-7B model as shown in Table E1.

Note that, after the model is trained with QLoRA-like method, it is not straightforward to fuse the high precision adapters $\boldsymbol{A}$ and $\boldsymbol{B}$ into the low-bit pretrained weights $\boldsymbol{W}_{\mathbb{Z}}$. Naively fusing them leads to a high quantization error, as demonstrated in Table E2. This issue persists even if the adapters $\boldsymbol{A}$ and $\boldsymbol{B}$ are quantized, as the resulting quantization grid of the product $\boldsymbol{AB}$ differs from that of $\boldsymbol{W_0}$. As mentioned above, QA-LoRA is the only work we are aware of that attempts to fuse auxiliary LoRA weights back into the frozen $\boldsymbol{W}_{\mathbb{Z}}$, however it achieves so by imposing constraints on quantization granularity.

We are inspired by LoRA-based methods to make QAT more memory- and runtime-efficient. Our goal is to design a method that is inference-efficient, *i.e.* where the low-rank adapters can be fused back into a low-bit integer matrix $\boldsymbol{W}_{\mathbb{Z}}$ without any loss of accuracy or perplexity. This way, we will not incur any additional inference overhead compared to PTQ, full-model QAT, or any other uniform affine quantization approaches. We summarize different trade-offs for the discussed techniques in Table 1.

## 4 METHOD

We now discuss the components of LR-QAT followed by a formal definition of LR-QAT. A schematic overview of our method is shown in Figure 1 (left).

**QAT with low-rank adapters** Let's recall how traditional QAT (Esser et al., 2020) works. Given a linear layer with a weight matrix $\boldsymbol{W} \in \mathbb{R}^{m \times k}$ and assuming $b$-bit symmetric uniform affine quantization, the quantization is simulated as follows:

$$\widehat{\boldsymbol{W}} := \mathbf{s} \cdot \text{clip}\left(\left\lfloor \frac{\boldsymbol{W}}{\mathbf{s}} \right\rceil; -2^{b-1}, 2^{b-1} - 1\right), \qquad (3)$$

where weights $\boldsymbol{W}$ are trainable parameters and the quantization scale $\mathbf{s}$ can be either fixed or also learned. To be able to backpropagate through round-to-nearest operation in (3), it is common to use *straight-through estimator* (STE, Bengio et al. 2013), where it is assumed that $\frac{\partial \lfloor t \rceil}{\partial t} = 1$. When applied to LLMs, it is straightforward to see that this procedure is very expensive: we have to learn a comparable number of parameters that was used for pretraining, leading to excessive memory usage.

To make this approach more practical we *freeze* the pretrained weights $\boldsymbol{W}$ (denote $\boldsymbol{W_0}$) and introduce low-rank adapters $\boldsymbol{A} \in \mathbb{R}^{m \times r}$, $\boldsymbol{B} \in \mathbb{R}^{r \times k}$, $r \ll \min\{m, k\}$. We have to be careful where exactly those adapters are placed. As discussed in Section 2, after the training is complete, we want $\boldsymbol{A}$ and $\boldsymbol{B}$ to be seamlessly integrated into a single $b$-bit integer matrix $\boldsymbol{W_\mathbb{Z}}$ without loss of accuracy to facilitate efficient inference. To accommodate that, we put the auxiliary matrices inside the rounding operator as follows

$$\widehat{\boldsymbol{W}} := \mathbf{s} \cdot \mathrm{clip}\left(\left\lfloor \frac{\boldsymbol{W_0}}{\mathbf{s}} + \frac{\alpha}{r} \boldsymbol{AB} \right\rceil; -2^{b-1}, 2^{b-1} - 1\right), \tag{4}$$

where we are using STE assumption for the rounding operation to compute the gradients of the loss w.r.t. $\boldsymbol{A}$, $\boldsymbol{B}$ and $\mathbf{s}$. We further employ a scaling factor $\alpha/r$ used in LoRA (Hu et al., 2021) to reduce the need to retune hyperparameters as we vary the rank $r$. After training is complete, (4) can be represented as regular fixed point tensor, $\widehat{\boldsymbol{W}} = \mathbf{s} \cdot \boldsymbol{W_\mathbb{Z}}$, without any extra effort or loss of accuracy and therefore enabling efficient inference without any extra overhead. Note that this is different to most of the literature, such as QLoRA (Dettmers et al., 2024), where adapters are placed outside of the quantization function (such as $\mathbf{y} = \widehat{\boldsymbol{W}}\mathbf{x} + \boldsymbol{AB}\mathbf{x}$) and are typically stored in higher precision formats such as BF16.

**Downcasting operator** The formulation (4) is already significantly more memory efficient compared to standard full-model QAT (3). We don't need to compute neither gradients w.r.t. weights $\boldsymbol{W}$ nor the respective first or second-order momentum terms for Adam-based optimizers, and only need to do so for the auxiliary matrices $\boldsymbol{A}$ and $\boldsymbol{B}$, which is noticeably more affordable provided $r \ll \min\{m, k\}$.

Given that the weight matrix $\boldsymbol{W_0}$ is frozen, the next natural step to further reduce the memory utilization is to store it in a lower-precision format. One could directly apply downcasting to $\boldsymbol{W_0}$ in (4). However, it's important to note that these weights are divided by the scale $\mathbf{s}$ during every forward pass. To ensure stable training, the scale generally needs to be stored in a high-precision format. Therefore, to simplify further, we propose the following variant of low-rank QAT:

$$\widehat{\boldsymbol{W}} := \mathbf{s} \cdot \mathrm{clip}\left(\left\lfloor \frac{\boldsymbol{W_0}}{\mathbf{s_0}} + \frac{\alpha}{r} \boldsymbol{AB} \right\rceil; -2^{b-1}, 2^{b-1} - 1\right), \tag{5}$$

where we use the initial scale[1] $\mathbf{s_0}$ instead of learned scale $\mathbf{s}$ inside the rounding operator, and the rest is the same as in (4). Now the entire fraction $\boldsymbol{W_0}/\mathbf{s_0}$ is fixed and we can store it in a lower-precision format. Note that the scale $\mathbf{s}$ outside of the clipping operator can still be learned. Empirically, we found that (5) performs consistently on par with or even slightly better compared to (4).

During training the pretrained weights are represented and stored as follows

$$\boldsymbol{\Phi_0} := \varphi\left(\frac{\boldsymbol{W_0}}{\mathbf{s_0}}\right), \tag{6}$$

where $\varphi(\cdot)$ is a *downcasting operator* that encapsulates a choice of different numeric formats or other preprocessing computations. In the simplest form, $\varphi(\cdot)$ would cast the input to one of pre-existing floating-point formats, such as FP16, BF16, FP8 etc.

Inspired by traditional fixed point quantization, we also explore integer representations for $\varphi(\cdot)$. Specifically, $\varphi(x) = \mathrm{clip}\left(\lfloor x \rceil, -2^{b-1}, 2^{b-1} - 1\right)$ corresponds to a standard $b$-bit integer quantization and can be stored as INT-$b$ number. We denote this approach $\varphi = $ INT-$b$ for brevity. In addition to that, in case of low-bit quantization ($b \leq 4$), which is a primary focus of this work, two INT-$b$ numbers can be *double-packed* into a single INT8 number, leading to further memory savings. This is helpful because most of the common deep learning frameworks like PyTorch, at the time of writing this paper, don't natively support low-bit formats such as INT4 yet.

---

[1]A frozen scale obtained after initial range estimation before the training begins.

Using $\varphi = \mathsf{INT}\text{-}b$ naturally leads to aggressive memory reduction by only keeping the integer part of (clipped) $\boldsymbol{W_0}/\mathbf{s_0}$. In our preliminary experiments, we found that this setting, combined with the standard initialization for $\boldsymbol{A}$ and $\boldsymbol{B}$ used in (Hu et al., 2021), did not work as well compared to $\varphi = \mathsf{BF16}$. This indicates the importance of keeping some information of the fractional part of $\boldsymbol{W_0}/\mathbf{s_0}$ and potentially the need for better initialization of auxiliary matrices.

We address this problem in two distinct ways: We adapt and experiment with a variant of SVD-based initialization for low-rank matrices $\boldsymbol{A}$, $\boldsymbol{B}$ proposed in (Li et al., 2023) before we apply a downcasting operator to $\boldsymbol{W_0}/\mathbf{s_0}$, to capture some of the information of it's fractional part. With this approach we can still employ a double-packing since we are still using $\varphi = \mathsf{INT}\text{-}b$.

Another way is to use $\mathsf{INT8}$ storage type, allocate $b$ bits to represent the integer part as before, but utilize the remaining $8-b$ bits for storing the approximate fractional part ($2 \leq b \leq 7$). In other words, we represent $\boldsymbol{\Phi_0}$ using fixed-point numbers. Assuming the rest of the computation is performed in $\mathsf{BF16}$, we define the downcasting and the corresponding upcasting operators as follows:

$$\begin{aligned}
\varphi(\mathbf{x}) &= \mathsf{INT8}\big(\big\lfloor 2^{8-b} \cdot \mathrm{clip}\left(\mathbf{x}; -2^{b-1}, 2^{b-1} - 1\right)\big\rceil\big), \\
\varphi^{-1}(\mathbf{x}) &= \mathsf{BF16}(\mathbf{x})/2^{8-b}.
\end{aligned} \tag{7}$$

A fixed-point number where $n$ bits are used for the integer part of the value and $m$ bits are used for the fractional part are commonly denoted (Oberstar, 2007) as Qn.m. For brevity, we will refer to (7) as $\varphi = \mathsf{Q}b.(8-b)$. In this work we will be mainly focusing on $b \in \{3, 4\}$, which corresponds to Q3.5 and Q4.4, respectively.

**Gradient checkpointing**  Note that both in the original LoRA paper (Hu et al., 2021) and in the related work like QLoRA (Dettmers et al., 2024), there is no need to compute the product $\boldsymbol{AB}$ explicitly. Instead, those matrices are multiplied with the activations $\mathbf{x}$ as $\boldsymbol{A}\left(\boldsymbol{B}\mathbf{x}\right)$. However, we have to compute a product $\boldsymbol{AB}$ in (5), and in the naïve implementation of our method, this product together with the results of some intermediate computations (e.g., after rounding and clipping) will be automatically kept in memory for the backward pass, leading to increased memory usage. To prevent this, we employ gradient checkpointing (Chen et al., 2016) on (5). In other words, we recompute the quantizer function in the backward pass, leading to a slight runtime overhead but avoiding significantly increased memory usage.

**LR-QAT**  Using the components described above, we define LR-QAT for a single layer with a (pretrained) weight matrix $\boldsymbol{W_0}$ as follows

$$\widehat{\boldsymbol{W}} := \mathbf{s} \cdot \mathrm{clip}\left(\left\lfloor \boldsymbol{\Phi_0} + \frac{\alpha}{r}\boldsymbol{AB}\right\rceil; -2^{b-1}, 2^{b-1} - 1\right), \tag{8}$$

where $\mathbf{s}$ – trainable or frozen quantization scale with the initial value of $\mathbf{s_0}$, $\boldsymbol{A}$, $\boldsymbol{B}$ – trainable rank $r$ auxiliary matrices, $\boldsymbol{\Phi_0} := \varphi(\boldsymbol{W_0}/\mathbf{s_0})$ – frozen representation of the original pretrained weights, and $\varphi$ is the downcasting operator. To avoid excessive memory allocation for the results of intermediate computations in (8) involving the product $\boldsymbol{AB}$, we apply checkpointing on $\widehat{\boldsymbol{W}}$. After the training is complete, low-rank adapters are naturally integrated into a single integer matrix $\boldsymbol{W_{\mathbb{Z}}} = \mathrm{clip}\left(\cdots\right)$ without loss of accuracy. Note, while we presented our method for symmetric quantization which is commonly used for weights (Nagel et al., 2021), it can equally be applied for asymmetric quantization by adding a zero offset $z$ outside the rounding operation as shown in (1).

## 5 EXPERIMENTS

We assess the effectiveness of LR-QAT by conducting experiments on LLaMA 7B (Touvron et al., 2023a), LLaMA-2 7B/13B (Touvron et al., 2023b), LLaMA-3 8B (AI@Meta, 2024), and Mistral-0.1 7B (Jiang et al., 2023). We first explore the impact of the choice of rank $r$, a downcasting operator $\varphi(\cdot)$, and the initialization of auxiliary matrices $\boldsymbol{A}$, $\boldsymbol{B}$. We then compare our method in terms of accuracy to standard full-model QAT, other baselines, and the related work. All detailed hyperparameters of our experiments are in Appendix A.

**Quantization**  We experiment with both weight-only and weight-activation quantization. The default settings are $\mathsf{INT4}\,/\,\mathsf{INT3}\,/\,\mathsf{INT2}$ per-channel (denoted 'pc') and group-wise weight quantization

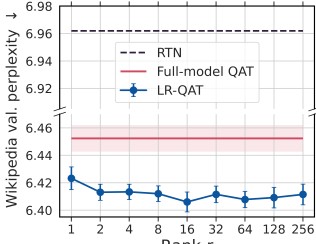

| $\varphi(\cdot)$ | dtype | $A$, $B$ init. | WikiText-2 ↓ | | Zero-shot acc. ↑ | |
|---|---|---|---|---|---|---|
| | | | W4 pc | W3 pc | W4 pc | W3 pc |
| FP32 | FP32 | LoRA | 5.69 | 6.21 | 69.28 | 66.62 |
| FP16 | FP32 | LoRA | $+0.00$ | $+\mathbf{0.01}$ | $-0.13$ | $-0.01$ |
| BF16 | FP32 | LoRA | $-\mathbf{0.01}$ | $+\mathbf{0.01}$ | $+0.11$ | $+\mathbf{0.45}$ |
| Q4.4 / Q3.5 | FP32 | LoRA | $-\mathbf{0.01}$ | $+\mathbf{0.01}$ | $+\mathbf{0.16}$ | $+0.31$ |
| **Q4.4 / Q3.5** | **BF16** | **LoRA** | $-\mathbf{0.01}$ | $+\mathbf{0.01}$ | $+0.15$ | $+0.31$ |
| INT-4 / INT-3 | FP32 | LoRA | $+0.02$ | $+20.5$ | $-0.04$ | $-22.8$ |
| INT-4 / INT-3 | FP32 | LoftQ ($T=1$) | $+0.28$ | $+0.18$ | $-0.67$ | $+0.26$ |
| INT-4 / INT-3 | FP32 | LoftQ ($T=64$) | $+0.40$ | $+1.37$ | $-1.40$ | $-2.01$ |

Figure 2 & Table 2: *Left*: The performance of LR-QAT ($\varphi = $ Q4.4) depending on the rank $r$ of auxiliary matrices $A$ and $B$ on LLaMA-2 7B with W4 per-channel quantization. We report mean and standard deviation over 5 runs with different random seeds. *Right*: The performance of LR-QAT applied to LLaMA-2 7B depending on the choice of downcasting operator $\varphi(\cdot)$, compute data type, and initialization method for low-rank auxiliary matrices. We report WikiText-2 test set perplexity, lower is better, and average zero-shot accuracy of 6 tasks, higher is better. Numbers marked in bold are the best results.

with a group size of 128 (denoted 'g128'). We use symmetric quantization, except the INT2 case, where we use asymmetric quantization (1), for a fair comparison with related work. We quantize all linear layers, except the classification head. In weight-activation quantization, defaults are INT4 per-channel weight and per-token activation quantization (Dettmers et al., 2022). Following OmniQuant (Shao et al., 2023), we quantize all inputs to matmuls with exception of the softmax output and additionally quantize the KV-cache as in LLM-QAT (Liu et al., 2023b).

**Datasets and training**   We apply our method to all linear layers in the attention blocks, both in self-attention and in the feed-forward network. We only train low-rank auxiliary matrices $A$, $B$ and the quantization parameters s and keep embedding layers, final classification head and RMSNorm parameters frozen. In the case of asymmetric weight quantization, a zero offset z is set during range estimation phase and kept frozen throughout training ($\mathbf{z} = \mathbf{z_0}$).

We train on a small subset of SlimPajama (Soboleva et al., 2023), which is an open-source dataset similar to the original one used for pretraining LLaMA models. In all experiments we train using batch size 32 and a maximum sequence length of 1024. For all weight-only and weight-activation quantization results, we train for $10^4$ steps. For ablation studies in Sections 5.1 and 5.2 we use shorter training of $10^3$ steps. We select hyperparameters based on the perplexity of a small subset of Wikipedia validation set (512 sequences).

**Evaluation**   Following the previous work (Frantar et al., 2022; Xiao et al., 2023; Shao et al., 2023; Liu et al., 2023b), we evaluate quantized models by reporting the perplexity of language generation on WikiText-2 (Merity et al., 2016), using a sequence length of 2048. We also report zero-shot accuracy on a set of common sense reasoning tasks including BoolQ (Clark et al., 2019), PIQA (Bisk et al., 2020), Winogrande (Sakaguchi et al., 2021), ARC (Clark et al., 2018), and HellaSwag (Zellers et al., 2019). For zero-shot evaluation, we use the LM Evaluation Harness framework (Gao et al., 2021).

**Baselines**   We compare with round-to-nearest quantization (RTN), where we set the ranges based on minimizing the $L^p$-norms between quantized and unquantized weights and report the best performing configuration. We also use that as initialization for LR-QAT. We investigate the impact of RTN initialization in Appendix D.

For weight-only quantization, we compare with GPTQ (Frantar et al., 2022), AWQ (Lin et al., 2023), OmniQuant (Shao et al., 2023), and BitDistiller (Du et al., 2024). To make a comparison with OmniQuant more complete, we generated and included additional baseline results using their public open-source code (see details in Appendix B). We also compare with our implementation of PEQA (Kim et al., 2024) and full-model QAT (LSQ) (Esser et al., 2020), where we follow the same experimental setup as for our method, together with the best RTN initialization, for a fair comparison.

For weight-activation quantization, we compare our method with RTN, SmoothQuant (Xiao et al., 2023), LLM-QAT (Liu et al., 2023b), Outlier Suppression+ (Wei et al., 2023), OmniQuant (Shao et al., 2023), and our implementation of PEQA (Kim et al., 2024). Following (Liu et al., 2023b), we

| Method | GPU mem., GB | Time/100 steps, sec | WikiText-2 ppl. ↓ | | Zero-shot acc. ↑ | |
|---|---|---|---|---|---|---|
| | | | W4 pc | W3 pc | W4 pc | W3 pc |
| Full-model QAT (LSQ) | 62.2 (98.5) | $3248^{\pm 7}$ | $5.77^{\pm 0.02}$ | $6.14^{\pm 0.01}$ | $68.96^{\pm 0.29}$ | $67.14^{\pm 0.13}$ |
| **LR-QAT (ours)** | **20.5** | $\mathbf{1522^{\pm 5}}$ | $\mathbf{5.66^{\pm 0.00}}$ | $\mathbf{6.13^{\pm 0.02}}$ | $\mathbf{69.72^{\pm 0.32}}$ | $\mathbf{67.70^{\pm 0.25}}$ |

Table 3: A comparison of the proposed method ($\varphi = $ Q4.4) with the full-model QAT on LLaMA-2 7B with W4 and W3 per-channel quantization. We report mean and standard deviation over 5 runs with different random seeds. We also report the maximum GPU memory with (without) gradient checkpointing and the training runtime on a Nvidia A100 80GB GPU.

compare to them in several different settings, where the weights, activations and KV cache values are quantized to different bitwidths (denoted as W-A-KV).

## 5.1 THE IMPACT OF RANK $r$

We investigate the effect of different values of rank $r$ of the auxiliary matrices $\boldsymbol{A}$ and $\boldsymbol{B}$ and present results in Figure 2. Increasing the rank from 1 to 32 leads to progressively slightly better performance, excluding one outlier. The fact that using $r > 32$ doesn't lead to further improvement in perplexity is likely because of the limited number of training steps we used for this experiment ($10^3$), and that more steps might be needed for the procedure to fully converge. Interestingly, a rank $r$ as small as 1 already performs really well. We hypothesize that this is the case because of the following. Even though $\text{rank}(\boldsymbol{AB}) = 1$, by applying a low-rank approximation inside the rounding and clipping operators in (8), this can overall lead to a high-rank perturbation to the original weights $\boldsymbol{\Phi_0}$ (in the integer domain). Finally, for all ranks we observe only a small standard deviation between 0.005 and 0.008 ppl., indicating the robustness of LR-QAT to a random initialization of $\boldsymbol{B}$. Going forward, we use $r = 32$ in all our experiments[2].

## 5.2 THE CHOICE OF THE DOWNCASTING OPERATOR $\varphi(\cdot)$ AND $\boldsymbol{A}$, $\boldsymbol{B}$ INITIALIZATION

We study the effect of several choices of the downcasting operators discussed in Section 4 and summarize results in Table 2. We can see that by going from FP32 to BF16, and finally to an 8-bit fixed-point representation of $\boldsymbol{\Phi_0}$, aside from memory savings we also maintain the same WikiText-2 perplexity and even slightly improve zero-shot accuracy. The latter is likely due to a slight regularization effect caused by the fact that we discard some of the information in the fractional part in $\boldsymbol{W_0}/\boldsymbol{s_0}$, some of which might be noise. One step further, however, while $\varphi = $ INT-$b$ still leads to a good model performance in the case of 4-bit weight quantization, it completely breaks for W3.

So far, we initialized matrices $\boldsymbol{A}$ and $\boldsymbol{B}$ following the procedure proposed in LoRA (Hu et al., 2021) where $\boldsymbol{B}$ is initialized to zero, and $\boldsymbol{A}$ is initialized randomly as in (He et al., 2015). We refer to this initialization scheme as 'LoRA'. We hypothesize that a poor performance of $\varphi = $ INT3 can be explained by the fact that we lose all the information in the fractional part of $\boldsymbol{W_0}/\boldsymbol{s_0}$ and that without that information it is difficult for low-rank approximation to learn. To address this, we adapt and experiment with a variant of SVD-based initialization proposed in LoftQ (Li et al., 2023). We see that using LoftQ initialization with $T = 1$ step recovers almost all the predictive performance compared to a fixed-point representation. Increasing number of LoftQ steps, or applying it to a 4-bit case, however, did not help.

Finally, when using the fixed point representation for $\boldsymbol{\Phi_0}$, we still maintain the same model performance by switching the compute data type[3] from FP32 to BF16, where the latter is what is commonly used for LLMs. Going forward, we use $\varphi = $ Q$b.(8 - b)$ with 'LoRA' initialization and BF16 compute data type.

## 5.3 COMPARISON WITH FULL-MODEL QAT

Finally, before presenting our main set of results, we compare our method with a standard full-model QAT (LSQ) (Esser et al., 2020). For full-model QAT, we follow the same training setup as for our method. We also tune the maximum value of the learning rate for $\boldsymbol{W}$ using the following search

---

[2]This amounts to only 1.2% of the total number of parameters for 7B LLaMA model.

[3]A data type used for activations, gradients, and frozen parameters.

| # Bits | Method | WikiText-2 perplexity ↓ | | | | | Avg. zero-shot accuracy ↑ | | | | |
|---|---|---|---|---|---|---|---|---|---|---|---|
| | | L1-7B | L2-7B | L2-13B | L3-8B | M-7B | L1-7B | L2-7B | L2-13B | L3-8B | M-7B |
| FP16 | | 5.68 | 5.47 | 4.88 | 6.14 | 5.25 | 69.68 | 70.47 | 73.18 | 74.22 | 75.69 |
| W4 pc | RTN | 6.33 | 6.14 | 5.21 | 7.53 | 5.91 | **68.51** | 68.88 | 71.73 | 72.19 | 73.44 |
| | GPTQ[§] | 6.13 | 5.83 | 5.13 | - | - | 64.95 | - | - | - | - |
| | AWQ | 6.08 | 6.15 | 5.12 | - | - | - | - | - | - | - |
| | OmniQuant[§] | **5.86** | 5.74 | **5.02** | 7.30 | 5.61 | 68.48 | 68.19 | 71.69 | 72.49 | 73.68 |
| | LSQ (our impl.) | 5.94 | 5.77 | ✕ | 6.87 | 5.73 | 68.37 | 68.96 | ✕ | 73.28 | 72.88 |
| | PEQA (our impl.) | **5.86** | **5.71** | **5.03** | 7.51 | 5.56 | **68.49** | 69.23 | 72.51 | 72.79 | 73.73 |
| | **LR-QAT (ours)** | **5.84** | **5.66** | **5.03** | 6.78 | 5.46 | **68.54** | **69.72** | **73.19** | 73.84 | **74.44** |
| W4 g128 | RTN | 6.05 | 5.78 | 5.04 | 6.96 | 5.49 | 68.93 | 69.75 | **72.94** | 72.30 | 75.07 |
| | GPTQ[§] | 5.85 | **5.61** | 4.98 | - | - | - | - | - | - | - |
| | AWQ | 5.81 | **5.62** | 4.97 | - | - | - | - | - | - | - |
| | OmniQuant[§] | **5.77** | 5.58 | 4.95 | 6.70 | **5.40** | **69.15** | 69.58 | 72.80 | 73.56 | **75.33** |
| | LSQ (our impl.) | **5.76** | **5.61** | ✕ | 6.58 | 5.67 | **69.17** | 69.68 | ✕ | 73.31 | 72.90 |
| | PEQA (our impl.) | **5.75** | 5.67 | 5.02 | 6.89 | 5.48 | **69.19** | 69.64 | 72.80 | 72.99 | 73.34 |
| | **LR-QAT (ours)** | **5.75** | **5.59** | 4.97 | 6.57 | 5.37 | **69.15** | **69.88** | 72.91 | 73.66 | 75.28 |
| W3 pc | RTN | 12.88 | 26.73 | 8.71 | 34.10 | 9.49 | 54.66 | 43.87 | 55.01 | 47.46 | 64.58 |
| | GPTQ[§] | 8.06 | 8.37 | 6.44 | - | - | - | - | - | - | - |
| | AWQ | 11.88 | 24.00 | 10.45 | - | - | - | - | - | - | - |
| | OmniQuant[§] | 6.49 | 6.58 | **5.58** | 15.91 | 7.13 | 66.40 | 63.94 | 70.20 | 56.83 | 67.40 |
| | LSQ (our impl.) | **6.29** | **6.14** | ✕ | **8.14** | **6.06** | 66.29 | 67.14 | ✕ | 69.58 | 71.61 |
| | PEQA (our impl.) | 6.56 | 6.45 | 5.73 | 26.20 | 6.51 | 65.75 | 65.44 | 69.81 | 51.05 | 71.02 |
| | **LR-QAT (ours)** | **6.27** | **6.13** | 5.54 | 8.12 | 6.03 | **66.60** | **67.70** | 71.22 | 70.46 | 71.87 |
| W3 g128 | RTN | 7.96 | 7.61 | 6.20 | 15.11 | 6.77 | 63.50 | 63.20 | 67.60 | 57.74 | 69.35 |
| | GPTQ[§] | 6.55 | 6.29 | 5.42 | - | - | - | - | - | - | - |
| | AWQ | 6.46 | 6.24 | 5.32 | - | - | - | - | - | - | - |
| | OmniQuant[§] | **6.15** | 6.03 | 5.28 | 8.81 | 5.86 | **66.77** | 67.52 | 70.97 | 66.28 | 73.06 |
| | BitDistiller[§] | - | 5.97 | 5.20 | - | - | - | ❖ | ❖ | - | - |
| | LSQ (our impl.) | **6.20** | **6.02** | ✕ | 8.08 | 5.90 | 66.53 | 68.36 | ✕ | 70.11 | 71.96 |
| | PEQA (our impl.) | 6.22 | 6.05 | 5.58 | 9.64 | **5.85** | 66.66 | 68.10 | 70.29 | 67.19 | 72.21 |
| | **LR-QAT (ours)** | **6.17** | **5.98** | 5.32 | 7.74 | 5.80 | **66.81** | **68.62** | 71.51 | 70.48 | 72.41 |
| W2 pc | RTN[§] | 4.9e3 | 5.2e3 | 5.2e3 | 6.4e4 | 6.8e3 | 37.92 | 36.52 | 36.27 | 36.80 | 36.59 |
| | GPTQ[§] | 2.1e3 | 7.7e3 | 2.1e3 | - | - | - | - | - | - | - |
| | OmniQuant[§] | 15.47 | 37.37 | 17.21 | 5.1e3 | 339 | 49.78 | 43.67 | 49.72 | 36.36 | 36.39 |
| | PEQA (our impl.)[§] | 8.24 | 9.34 | 7.51 | 14.47 | 8.08 | 59.83 | 57.40 | 62.29 | 57.72 | 64.48 |
| | **LR-QAT (ours)[§]** | **7.99** | **8.51** | **7.16** | **12.52** | **7.98** | **61.77** | **60.03** | **65.28** | **58.49** | **65.11** |
| W2 g128 | RTN[§] | 708 | 2.5e3 | 115.6 | 1.4e4 | 369 | 39.74 | 37.94 | 41.12 | 36.97 | 41.30 |
| | GPTQ[§] | 44.01 | 36.77 | 28.14 | - | - | - | - | - | - | - |
| | AWQ | 2.6e5 | 2.2e5 | 1.2e5 | - | - | - | - | - | - | - |
| | OmniQuant[§] | 9.72 | 11.06 | 8.26 | 327 | 16.06 | 54.31 | 52.00 | 57.16 | 37.70 | 50.46 |
| | BitDistiller[§] | - | 8.08 | 6.78 | - | - | - | ❖ | ❖ | - | - |
| | PEQA (our impl.)[§] | 8.12 | 7.87 | 6.83 | 12.39 | 8.05 | 60.69 | 60.74 | 64.74 | 57.81 | 64.65 |
| | **LR-QAT (ours)[§]** | **7.86** | **7.62** | **6.57** | **11.09** | **7.92** | **61.60** | **61.70** | **66.75** | **60.46** | **65.27** |

Table 4: **Weight-only quantization results for LLaMA and Mistral models**. We report WikiText-2 test set perplexity (lower is better) and average zero-shot accuracy (higher is better). Models marked 'L1'/'L2'/'L3', and 'M' denote LLaMA-1/2/3 and Mistral, respectively. Numbers marked in bold are SOTA or on par (within 0.05). [§]Uses asymmetric weight quantization. ✕ denotes out of memory. ❖ denotes that method reports results using a different set of metrics for zero-shot evaluation, see Table D3 for a fair comparison.

space {1e-5, **5e-5**, 1e-4, 5e-4, 1e-3} and select the best configuration based on Wikipedia validation perplexity. Note that we use the same learning rate for $s$ for both full-model QAT and our method.

As we can see in Table 3, training with our method leads to on par or better predictive performance at a significantly lower memory usage and training runtime compared to full-model QAT. From Figure 1, right, we can see that LR-QAT drastically reduces memory requirements for gradients and optimizer state and halves memory requirements for weights. Finally, thanks to gradient checkpointing, it further decreases memory usage by not storing results of intermediate computations within quantization function. We include results for other models and bitwidths in Table 4. A more detailed runtime comparison can also be found in Appendix F.

## 5.4 MAIN RESULTS

**Weight-only quantization** We summarize our results in Table 4. As we can see, in almost most cases LR-QAT outperforms or is on par with prior weight-only quantization methods across various LLM families and quantization settings, including both per-channel and group-wise quantization.

| # Bits (W-A-KV) | Method | WikiText-2 perplexity ↓ | | | Avg. zero-shot accuracy ↑ | | |
|---|---|---|---|---|---|---|---|
| | | L1-7B | L2-7B | L2-13B | L1-7B | L2-7B | L2-13B |
| FP16 | | 5.68 | 5.47 | 4.88 | 69.68 | 70.47 | 73.18 |
| 4-8-8 | RTN | 6.88 | 6.17 | 5.23 | 65.83 | 68.55 | 71.54 |
| | SmoothQuant | 13.7[*] | - | - | 65.17 | - | - |
| | LLM-QAT | 11.2[*] | - | - | 68.18 | - | - |
| | PEQA (our impl.) | 5.89 | 5.72 | 5.08 | 68.53 | 69.11 | 72.49 |
| | **LR-QAT (ours)** | **5.85** | **5.67** | **5.04** | **68.58** | **69.32** | **73.18** |
| 4-8-4 | RTN | 7.66 | 6.85 | 5.78 | 62.78 | 65.16 | 67.06 |
| | SmoothQuant | 163.6[*] | - | - | 45.35 | - | - |
| | LLM-QAT | 11.6[*] | - | - | 64.75 | - | - |
| | PEQA (our impl.) | 6.15 | 6.03 | 5.27 | 66.54 | 67.56 | 71.35 |
| | **LR-QAT (ours)** | **6.07** | **5.90** | **5.24** | **67.16** | **69.84** | **71.65** |
| 4-4-4 | RTN | 17.75 | 18.98 | 11.37 | 49.60 | 51.75 | 55.07 |
| | SmoothQuant | 25.25 | 83.12 | 35.88 | 38.42 | - | - |
| | LLM-QAT | - | - | - | 41.27 | - | - |
| | LLM-QAT + SQ | - | - | - | 46.43 | - | - |
| | Outlier Suppression+ | - | - | - | 48.43 | - | - |
| | OmniQuant[§] | 11.26 | 14.26 | 12.30 | 52.65 | - | - |
| | PEQA (our impl.) | 8.60 | 8.72 | 7.23 | 58.39 | 57.93 | 62.26 |
| | **LR-QAT (ours)** | **8.47** | **8.46** | **7.15** | **59.00** | **58.98** | **62.65** |

Table 5: **Weight and activation quantization results for LLaMA-1/2** (denoted 'L1'/'L2', respectively). We report WikiText-2 test set perplexity and zero-shot accuracy of 6 tasks. Numbers marked in bold are SOTA. [§]Uses asymmetric weight quantization. [*]Uses a maximum sequence length of 1024 for evaluation.

In the case of extremely low-bitwidth regime (W2), our method consistently and significantly outperforms related work, across all settings.

In a few cases our method did not outpeform OmniQuant and BitDistiller. However, both methods employ asymmetric quantization which provides extra degrees of freedom compared to symmetric quantization, which are very helpful in the case of low-bit quantization. In practice, however, symmetric weight quantization yields more efficient inference (Nagel et al., 2021). Additionally, techniques like OmniQuant and related techniques are orthogonal to our method and can be used as as initialization of LR-QAT.

**Weight-activation quantization**   We present our results for weight-activation quantization applied to LLaMA-1/2 models in Table 5. LR-QAT consistently outperforms all PTQ and QAT baselines, across all model families and the bitwidth settings. In addition to that, as we decrease the activation bitwidths, the improvement in model performance compared to prior work becomes more pronounced.

This indicates LR-QAT's versatility, being readily applicable not only to weight-only quantization but also weight-activation quantization, a setting that allows for a very efficient inference using fixed-point arithmetic. Further, our method can still be combined with most of the related PTQ methods including OmniQuant that shift the difficulty of activation quantization to weight quantization, and will likely lead to even better results.

# 6 CONCLUSIONS

In this paper we propose LR-QAT, a lightweight and memory-efficient QAT algorithm for LLMs which enables training a 7B LLM on a single consumer grade GPU with 24GB of memory. Inspired by PEFT methods, we introduce a low-rank reparameterization that is aware of the quantization grid. We further reduce the memory requirements by introducing a downcasting operator involving fixed-point or double-packed integers, and applying checkpointing. In almost all cases, our method outperforms common PTQ approaches and reaches the same model performance as full-model QAT at the fraction of its memory usage.

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

## A EXPERIMENTAL DETAILS

In this section, we list the details related to hyperparameters and other settings used in our experiments. If not stated otherwise, the standard hyperparameters that we use are the one shown in Table A1.

| Hyperparameter | Value / Search space |
|---|---|
| Optimizer | AdamW |
| Learning rate for $A$, $B$ (INT4 / INT3 / INT2) | $\{10^{-5}, 10^{-4}, 10^{-3}, 10^{-2}\}$ |
| Learning rate for $s$ (INT4 / INT3) | $\{0^{*}, 10^{-5}\}$ |
| Learning rate for $s$ (INT2) | $\{10^{-5}, 10^{-4}\}$ |
| Learning rate for $z$ (INT2 only) | $0^{*}$ |
| Learning rate for $W$ (full-model QAT only) | $\{1, 5, 10, 50, 100\} \cdot 10^{-5}$ |
| Learning rate schedule for $A$, $B$ | `linear` (with warmup) |
| Learning rate schedule for $s$ | `linear` (with warmup) |
| Learning rate schedule for $W$ (full-model QAT only) | `linear` (with warmup) |
| Weight decay for $A$, $B$ | 0 |
| Weight decay for $s$ | 0 |
| Weight decay for $W$ (full-model QAT only) | 0.1 |
| Adam $(\beta_1, \beta_2)$ | $(0.9, 0.95)$ |
| Training steps | $10^4$ |
| Warmup steps | 10% of Training steps |
| Batch size | 32 |
| Maximum sequence length (during training) | 1024 |
| $L^2$-norm gradient clipping (maximum norm) | 1.0 |
| $\alpha$ in (8) | 1.0 |

Table A1: Common hyperparameters used for experiments. *Is equivalent to freezing the quantization scale / zero offset obtained after initial range estimation ($s = s_0$, $z = z_0$).

**Quantization** We experiment with both weight-only and weight-activation quantization. The default settings are INT4 / INT3 / INT2 per-channel (denoted 'pc') and group-wise weight quantization with a group size of 128 (denoted 'g128'). We use symmetric quantization, except the INT2 case, where we use asymmetric quantization (1), for a fair comparison with related work. We quantize all linear layers, except the classification head. In weight-activation quantization, defaults are INT4 per-channel weight and per-token activation quantization (Dettmers et al., 2022). Following OmniQuant (Shao et al., 2023), we quantize all inputs to matmuls with exception of the softmax output and additionally quantize the KV-cache as in LLM-QAT (Liu et al., 2023b).

**Libraries** We implement our method in PyTorch (Paszke et al., 2019) and use training and evaluation pipelines from HuggingFace libraries Gugger et al. (2022); Lhoest et al. (2021); Wolf et al. (2020). For zero-shot evaluation, we use the LM Evaluation Harness framework (Gao et al., 2021). Specifically, we use `lm_eval` v0.4.2 and report `acc_norm` for tasks where it's available (PIQA, ARC-e, ARC-c, HellaSwag) and otherwise `acc` (BoolQ and Winogrande).

**Datasets and training** To optimize the learnable parameters, we use AdamW optimizer (Loshchilov and Hutter, 2017) with weight decay set to zero, $(\beta_1, \beta_2) = (0.9, 0.95)$ and linear learning rate warm up over the first 10% steps, following by a linear decay to zero by the end of training. We use a separate maximum learning rate for quantization scales and for low-rank adapters, which are tuned depending on the experiment.

We apply our methods to all linear layers in the attention blocks (both in self-attention and in the feed-forward network). We only train low-rank auxiliary matrices $A$, $B$ and the quantization parameters $s$. Specifically, we freeze embedding layers, the final classification heads and RMSNorm parameters. In the case of asymmetric weight quantization, a zero offset $z$ is set during range estimation phase and kept frozen throughout training.

We train on a small subset of SlimPajama (Soboleva et al., 2023), which is a close open-source replica of the dataset used for pre-training LLaMA models. We select hyperparameters based on the

perplexity of a small subset of Wikipedia validation set[4] (512 sequences). For all weight-only and weight-activation quantization results, including the comparison with full-model QAT in Section 5.3, we train for $10^4$ steps. For ablation studies in Sections 5.1 and 5.2 we use shorter training of $10^3$ steps. Since the full-model QAT experiment requires more than 80GB of GPU memory, we apply checkpointing on the quantization function $\widehat{W}$ to be able to run the experiment on an Nvidia A100 80GB GPU. We have also experimented with CPU optimizer state offloading, but that turned out to be significantly slower, see details in Table F1. The rest of the hyperparameters and their search spaces are listed in Table A1.

**PTQ initialization**  We compare with vanilla round-to-nearest quantization (RTN), where we explore several choices of range setting and report the best one based on Wikipedia validation set perplexity, and also use that as initialization for our method. Specifically, we experimented with min-max range estimator and with $L^p$-norm range estimator with the following values for $p$: $\{2.0, 2.4, 3.0, 3.5, 4.0, 5.0\}$.

## B  RESULTS SOURCES

In this study, we present a thorough comparison of our method against existing PTQ and QAT techniques. The results we discuss from their respective official publications, other scholarly articles, or are obtained from our reproduction. We carefully document the source of the results for each method as follows:

- *RTN* (round-to-nearest): our evaluation.
- *GPTQ*: as reported by OmniQuant (Shao et al., 2023).
- *AWQ*: as reported by OmniQuant (Shao et al., 2023).
- *OmniQuant*: as reported by OmniQuant (Shao et al., 2023) and additional results obtained by using their public open-sourced code base (see details below).
- *SmoothQuant*: as reported by LLM-QAT (Liu et al., 2023b) and OmniQuant (Shao et al., 2023).
- *LLM-QAT*: as reported by LLM-QAT (Liu et al., 2023b) and OmniQuant (Shao et al., 2023).
- *LLM-QAT + SQ* (LLM-QAT w/ SmoothQuant initialization): as reported by OmniQuant (Shao et al., 2023).
- *OS+* (Outlier Suppression+): as reported by OmniQuant (Shao et al., 2023).
- *BitDistiller*: as reported by BitDistiller (Du et al., 2024).
- *PEQA*: our implementation
- *LSQ* (Full-model QAT): our implementation

**Extra OmniQuant baseline results**  OmniQuant, being the strongest PTQ baseline that consistently outperforms other PTQ techniques (GPTQ, AWQ, RTN), only reports WikiText-2 perplexity results for LLaMA-1/2 models. To make the comparison more complete, we used their public open-sourced code base[5], to generate the missing results for LLaMA-3, Mistral, and zero-shot accuracy numbers for LLaMA-1/2.

We used the provided checkpoints by the authors for LLaMA-1/2 model families, which we exported and evaluated using our pipeline to make sure the consistent use of libraries and their versions, specifically `lm_eval` v0.4.2. We managed to closely match the WikiText-2 perplexity numbers reported by OmniQuant.

For LLaMA-3 and Mistral, we obtained the results using the provided run commands in the repo. Specifically, we followed the same experimental setup as OmniQuant: 20 epochs (40 epochs for W2) over 128 WikiText-2 sequences of length 2048. The rest of hyperparameters (including learning rate) were not changed. As a sanity check, we managed to closely match numbers for LLaMA-1 in Table A9 of (Shao et al., 2023).

---

[4]Specifically, we use the English subset of Wiki40b, `https://huggingface.co/datasets/wiki40b`, that contains cleaned-up text of English Wikipedia and training/validation splits.

[5]`https://github.com/OpenGVLab/OmniQuant`

## C  DETAILED ZERO-SHOT RESULTS

In this section, we provide a detailed breakdown of the task-specific accuracy numbers for the main set of results.

For weight-only quantization:

- LLaMA-1 7B in Table C1,
- LLaMA-2 7B in Table C2,
- LLaMA-2 13B in Table C3,
- LLaMA-3 8B in Table C4,
- Mistral 7B in Table C5,

and for weight-activation quantization in Table C6.

| # Bits | Method | BoolQ | PIQA | Winogrande | ARC-e | ARC-c | HellaSwag | Avg. |
|--------|--------|-------|------|-----------|-------|-------|-----------|------|
| FP16 | | 75.05 | 79.16 | 70.01 | 72.85 | 44.80 | 76.21 | 69.68 |
| W4 pc | RTN | 73.18 | 78.78 | 69.14 | 71.38 | 44.37 | 74.22 | 68.51 |
| | GPTQ[§] | 67.70 | 76.00 | 66.70 | 66.90 | 43.00 | 69.40 | 64.95 |
| | OmniQuant[§] | 74.68 | 79.00 | 68.59 | 71.34 | 42.58 | 74.73 | 68.48 |
| | LSQ (our impl.) | 73.91 | 78.24 | 69.22 | 70.84 | 43.26 | 74.74 | 68.37 |
| | PEQA (our impl.) | 74.71 | 78.29 | 70.09 | 70.33 | 42.24 | 75.27 | 68.49 |
| | **LR-QAT (ours)** | **74.13** | **78.29** | **70.01** | **71.21** | **42.41** | **75.16** | **68.54** |
| W4 g128 | RTN | 74.77 | 78.51 | 70.64 | 71.30 | 43.60 | 74.74 | 68.93 |
| | OmniQuant[§] | 75.29 | 78.40 | 69.38 | 72.69 | 43.94 | 75.23 | 69.15 |
| | LSQ (our impl.) | 75.90 | 79.22 | 70.01 | 71.42 | 43.34 | 75.15 | 69.17 |
| | PEQA (our impl.) | 75.75 | 79.17 | 70.17 | 70.75 | 43.60 | 75.71 | 69.19 |
| | **LR-QAT (ours)** | **75.29** | **78.62** | **69.61** | **71.59** | **44.11** | **75.67** | **69.15** |
| W3 pc | RTN | 58.93 | 70.40 | 55.72 | 55.01 | 32.17 | 55.75 | 54.66 |
| | OmniQuant[§] | 73.18 | 77.09 | 67.17 | 70.12 | 39.76 | 71.06 | 66.40 |
| | LSQ (our impl.) | 71.35 | 77.97 | 68.82 | 66.33 | 40.10 | 73.14 | 66.29 |
| | PEQA (our impl.) | 72.69 | 77.15 | 65.90 | 68.27 | 38.91 | 71.60 | 65.75 |
| | **LR-QAT (ours)** | **73.24** | **78.18** | **67.40** | **67.47** | **40.53** | **72.77** | **66.60** |
| W3 g128 | RTN | 69.48 | 76.33 | 64.40 | 64.44 | 38.65 | 67.67 | 63.50 |
| | OmniQuant[§] | 72.45 | 78.73 | 66.93 | 68.77 | 41.21 | 72.53 | 66.77 |
| | LSQ (our impl.) | 71.04 | 77.97 | 68.11 | 68.27 | 40.44 | 73.37 | 66.53 |
| | PEQA (our impl.) | 71.65 | 78.24 | 68.51 | 68.18 | 40.10 | 73.30 | 66.66 |
| | **LR-QAT (ours)** | **72.84** | **78.02** | **67.40** | **68.52** | **41.04** | **73.04** | **66.81** |
| W2 pc | RTN[§] | 43.24 | 52.61 | 51.62 | 27.48 | 26.28 | 26.26 | 37.92 |
| | OmniQuant[§] | 61.41 | 64.53 | 54.30 | 46.80 | 27.82 | 43.82 | 49.78 |
| | PEQA (our impl.)[§] | 67.19 | 74.32 | 61.80 | 59.97 | 32.42 | 63.28 | 59.83 |
| | **LR-QAT (ours)[§]** | **68.07** | **74.27** | **64.80** | **61.03** | **36.95** | **65.48** | **61.77** |
| W2 g128 | RTN[§] | 41.16 | 58.05 | 50.20 | 34.97 | 23.12 | 30.93 | 39.74 |
| | OmniQuant[§] | 62.94 | 68.17 | 56.75 | 53.24 | 30.55 | 54.20 | 54.31 |
| | PEQA (our impl.)[§] | 67.40 | 74.21 | 62.04 | 60.98 | 35.32 | 64.16 | 60.69 |
| | **LR-QAT (ours)[§]** | **67.65** | **74.86** | **61.80** | **62.37** | **37.03** | **65.90** | **61.60** |

Table C1: **LM-eval weight-only quantization results for LLaMA-1 7B**. We report zero-shot accuracy of 6 tasks (higher is better). [§]Uses asymmetric weight quantization.

| # Bits | Method | BoolQ | PIQA | Winogrande | ARC-e | ARC-c | HellaSwag | Avg. |
|---|---|---|---|---|---|---|---|---|
| FP16 | | 77.74 | 79.11 | 69.14 | 74.58 | 46.25 | 75.98 | 70.47 |
| W4 pc | RTN | 76.36 | 78.07 | 68.19 | 71.21 | 44.80 | 74.65 | 68.88 |
| | OmniQuant§ | 74.37 | 78.45 | 68.90 | 71.04 | 42.49 | 73.85 | 68.19 |
| | LSQ (our impl.) | 76.69 | 78.15 | 68.12 | 71.84 | 44.18 | 74.76 | 68.96 |
| | PEQA (our impl.) | 77.49 | 78.24 | 69.61 | 70.96 | 43.52 | 75.54 | 69.23 |
| | **LR-QAT (ours)** | **77.41** | **78.56** | **69.42** | **72.80** | **44.66** | **75.45** | **69.72** |
| W4 g128 | RTN | 76.76 | 78.18 | 69.77 | 72.60 | 45.73 | 75.43 | 69.75 |
| | OmniQuant§ | 77.19 | 79.05 | 68.11 | 73.70 | 44.54 | 74.90 | 69.58 |
| | LSQ (our impl.) | 77.28 | 78.45 | 69.61 | 72.18 | 44.97 | 75.61 | 69.68 |
| | PEQA (our impl.) | 76.88 | 78.89 | 69.85 | 72.18 | 44.11 | 75.95 | 69.64 |
| | **LR-QAT (ours)** | **76.73** | **78.62** | **70.48** | **72.85** | **44.97** | **75.62** | **69.88** |
| W3 pc | RTN | 46.27 | 60.28 | 54.85 | 38.05 | 23.29 | 40.47 | 43.87 |
| | OmniQuant§ | 68.72 | 74.43 | 66.30 | 65.57 | 38.74 | 69.91 | 63.94 |
| | LSQ (our impl.) | 74.39 | 77.91 | 66.85 | 69.15 | 41.53 | 73.04 | 67.14 |
| | PEQA (our impl.) | 71.62 | 76.82 | 66.14 | 65.66 | 39.76 | 72.63 | 65.44 |
| | **LR-QAT (ours)** | **75.08** | **77.73** | **67.50** | **69.97** | **42.70** | **73.24** | **67.70** |
| W3 g128 | RTN | 66.42 | 75.57 | 65.19 | 64.90 | 38.14 | 68.96 | 63.20 |
| | OmniQuant§ | 72.26 | 78.18 | 68.11 | 71.30 | 42.24 | 73.00 | 67.52 |
| | LSQ (our impl.) | 74.68 | 77.80 | 68.35 | 71.76 | 44.11 | 73.46 | 68.36 |
| | PEQA (our impl.) | 75.38 | 77.97 | 68.59 | 70.62 | 42.32 | 73.74 | 68.10 |
| | **LR-QAT (ours)** | **76.61** | **77.31** | **68.98** | **72.05** | **42.58** | **74.20** | **68.62** |
| W2 pc | RTN§ | 39.57 | 50.71 | 50.20 | 26.73 | 26.28 | 25.63 | 36.52 |
| | OmniQuant§ | 58.62 | 58.05 | 51.78 | 36.11 | 24.83 | 32.65 | 43.67 |
| | PEQA (our impl.)§ | 64.98 | 71.33 | 58.09 | 55.22 | 33.11 | 61.66 | 57.40 |
| | **LR-QAT (ours)§** | **69.88** | **73.07** | **62.90** | **56.57** | **34.04** | **63.71** | **60.03** |
| W2 g128 | RTN§ | 39.08 | 55.93 | 50.99 | 29.84 | 24.06 | 27.71 | 37.94 |
| | OmniQuant§ | 61.90 | 67.03 | 56.43 | 46.76 | 28.84 | 51.07 | 52.00 |
| | PEQA (our impl.)§ | 69.51 | 73.83 | 63.46 | 58.21 | 34.90 | 64.53 | 60.74 |
| | **LR-QAT (ours)§** | **70.18** | **73.83** | **63.69** | **60.27** | **34.98** | **67.24** | **61.70** |

Table C2: **LM-eval weight-only quantization results for LLaMA-2 7B**. We report zero-shot accuracy of 6 tasks (higher is better). §Uses asymmetric weight quantization.

| # Bits | Method | BoolQ | PIQA | Winogrande | ARC-e | ARC-c | HellaSwag | Avg. |
|---|---|---|---|---|---|---|---|---|
| FP16 | | 80.55 | 80.52 | 72.22 | 77.44 | 48.98 | 79.38 | 73.18 |
| W4 pc | RTN | 79.30 | 79.71 | 70.01 | 75.51 | 48.89 | 76.96 | 71.73 |
| | OmniQuant§ | 77.89 | 80.36 | 70.40 | 75.46 | 48.29 | 77.75 | 71.69 |
| | PEQA (our impl.) | 78.99 | 80.14 | 71.27 | 76.43 | 48.98 | 79.24 | 72.51 |
| | **LR-QAT (ours)** | **80.15** | **80.09** | **72.06** | **77.65** | **49.91** | **79.28** | **73.19** |
| W4 g128 | RTN | 81.10 | 79.82 | 72.38 | 76.73 | 49.06 | 78.52 | 72.94 |
| | OmniQuant§ | 80.37 | 79.82 | 72.45 | 76.68 | 49.15 | 78.32 | 72.80 |
| | PEQA (our impl.) | 80.28 | 80.63 | 71.74 | 76.14 | 48.38 | 79.62 | 72.80 |
| | **LR-QAT (ours)** | **80.73** | **80.30** | **71.74** | **76.14** | **49.06** | **79.51** | **72.91** |
| W3 pc | RTN | 55.05 | 71.06 | 54.22 | 56.19 | 32.25 | 61.27 | 55.01 |
| | OmniQuant§ | 77.74 | 79.22 | 69.06 | 74.12 | 45.65 | 75.39 | 70.20 |
| | PEQA (our impl.) | 74.28 | 78.67 | 69.06 | 74.87 | 45.99 | 76.00 | 69.81 |
| | **LR-QAT (ours)** | **78.62** | **79.49** | **72.61** | **73.99** | **45.56** | **77.05** | **71.22** |
| W3 g128 | RTN | 74.65 | 76.93 | 69.14 | 70.16 | 42.66 | 72.06 | 67.60 |
| | OmniQuant§ | 78.17 | 79.49 | 70.09 | 74.49 | 47.01 | 76.60 | 70.97 |
| | PEQA (our impl.) | 78.56 | 78.73 | 69.85 | 73.61 | 44.28 | 76.69 | 70.29 |
| | **LR-QAT (ours)** | **79.79** | **79.60** | **70.64** | **74.24** | **46.76** | **78.00** | **71.51** |
| W2 pc | RTN§ | 38.35 | 48.97 | 48.54 | 27.78 | 27.99 | 25.97 | 36.27 |
| | OmniQuant§ | 57.03 | 63.33 | 52.25 | 46.04 | 28.50 | 51.18 | 49.72 |
| | PEQA (our impl.)§ | 71.19 | 74.48 | 59.67 | 62.96 | 37.29 | 68.14 | 62.29 |
| | **LR-QAT (ours)§** | **72.60** | **76.61** | **66.22** | **66.67** | **39.76** | **69.79** | **65.28** |
| W2 g128 | RTN§ | 50.12 | 57.29 | 50.36 | 34.43 | 22.18 | 32.35 | 41.12 |
| | OmniQuant§ | 64.83 | 70.51 | 57.85 | 57.07 | 33.53 | 59.17 | 57.16 |
| | PEQA (our impl.)§ | 70.34 | 76.88 | 66.85 | 64.94 | 38.74 | 70.66 | 64.74 |
| | **LR-QAT (ours)§** | **75.47** | **77.86** | **65.98** | **67.63** | **41.04** | **72.50** | **66.75** |

Table C3: **LM-eval weight-only quantization results for LLaMA-2 13B**. We report zero-shot accuracy of 6 tasks (higher is better). §Uses asymmetric weight quantization.

| # Bits | Method | BoolQ | PIQA | Winogrande | ARC-e | ARC-c | HellaSwag | Avg. |
|---|---|---|---|---|---|---|---|---|
| FP16 | | 81.44 | 80.79 | 72.85 | 77.74 | 53.33 | 79.16 | 74.22 |
| W4 pc | RTN | 79.02 | 78.56 | 72.85 | 75.97 | 49.32 | 77.44 | 72.19 |
| | OmniQuant[§] | 79.17 | 78.89 | 72.77 | 76.56 | 50.34 | 77.20 | 72.49 |
| | LSQ (our impl.) | 80.58 | 80.03 | 72.69 | 78.24 | 50.17 | 77.94 | 73.28 |
| | PEQA (our impl.) | 79.57 | 78.67 | 72.93 | 77.19 | 51.11 | 77.25 | 72.79 |
| | **LR-QAT (ours)** | **81.62** | **79.98** | **72.85** | **78.32** | **52.05** | **78.19** | **73.84** |
| W4 g128 | RTN | 79.48 | 79.27 | 73.56 | 75.08 | 48.81 | 77.61 | 72.30 |
| | OmniQuant[§] | 79.79 | 80.36 | 73.24 | 78.37 | 51.45 | 78.16 | 73.56 |
| | LSQ (our impl.) | 80.24 | 80.36 | 73.01 | 77.23 | 50.43 | 78.60 | 73.31 |
| | PEQA (our impl.) | 80.98 | 80.14 | 72.61 | 76.18 | 49.57 | 78.45 | 72.99 |
| | **LR-QAT (ours)** | **80.40** | **80.90** | **73.48** | **77.44** | **51.11** | **78.60** | **73.66** |
| W3 pc | RTN | 58.65 | 61.75 | 56.04 | 39.60 | 23.81 | 44.91 | 47.46 |
| | OmniQuant[§] | 66.27 | 70.35 | 59.19 | 52.90 | 31.48 | 60.75 | 56.83 |
| | LSQ (our impl.) | 76.42 | 78.24 | 70.01 | 72.31 | 45.22 | 75.26 | 69.58 |
| | PEQA (our impl.) | 63.18 | 64.74 | 57.62 | 43.39 | 26.88 | 50.48 | 51.05 |
| | **LR-QAT (ours)** | **77.46** | **78.51** | **69.85** | **74.83** | **47.35** | **74.73** | **70.46** |
| W3 g128 | RTN | 65.47 | 68.39 | 65.19 | 54.00 | 33.45 | 59.96 | 57.74 |
| | OmniQuant[§] | 70.21 | 77.48 | 69.46 | 65.57 | 41.98 | 72.98 | 66.28 |
| | LSQ (our impl.) | 75.50 | 78.78 | 69.69 | 73.57 | 48.55 | 74.55 | 70.11 |
| | PEQA (our impl.) | 72.26 | 76.06 | 67.80 | 69.02 | 46.08 | 71.89 | 67.19 |
| | **LR-QAT (ours)** | **72.97** | **79.38** | **71.67** | **74.37** | **49.06** | **75.44** | **70.48** |
| W2 pc | RTN[§] | 44.89 | 48.97 | 47.51 | 25.29 | 27.73 | 26.41 | 36.80 |
| | OmniQuant[§] | 37.83 | 51.03 | 49.49 | 28.37 | 24.40 | 27.03 | 36.36 |
| | PEQA (our impl.)[§] | 62.54 | 72.96 | 59.51 | 56.82 | 33.70 | 60.77 | 57.72 |
| | **LR-QAT (ours)[§]** | **65.81** | **72.31** | **64.72** | **53.54** | **33.45** | **61.11** | **58.49** |
| W2 g128 | RTN[§] | 38.47 | 53.32 | 51.78 | 28.75 | 22.70 | 26.81 | 36.97 |
| | OmniQuant[§] | 45.14 | 51.74 | 49.96 | 28.24 | 22.61 | 28.51 | 37.70 |
| | PEQA (our impl.)[§] | 58.53 | 72.52 | 61.96 | 57.41 | 35.32 | 61.14 | 57.81 |
| | **LR-QAT (ours)[§]** | **67.98** | **73.83** | **62.83** | **56.94** | **36.43** | **64.77** | **60.46** |

Table C4: **LM-eval weight-only quantization results for LLaMA-3 8B**. We report zero-shot accuracy of 6 tasks (higher is better). [§]Uses asymmetric weight quantization.

| # Bits | Method | BoolQ | PIQA | Winogrande | ARC-e | ARC-c | HellaSwag | Avg. |
|---|---|---|---|---|---|---|---|---|
| FP16 | | 83.58 | 82.10 | 73.88 | 79.59 | 53.92 | 81.07 | 75.69 |
| W4 pc | RTN | 81.22 | 80.63 | 72.53 | 76.77 | 50.09 | 79.41 | 73.44 |
| | OmniQuant[§] | 81.35 | 81.50 | 72.45 | 76.26 | 50.17 | 80.35 | 73.68 |
| | LSQ (our impl.) | 81.96 | 80.41 | 73.09 | 75.08 | 48.55 | 78.19 | 72.88 |
| | PEQA (our impl.) | 81.80 | 81.12 | 72.61 | 77.23 | 50.17 | 79.43 | 73.73 |
| | **LR-QAT (ours)** | **81.99** | **81.28** | **73.56** | **78.20** | **51.02** | **80.57** | **74.44** |
| W4 g128 | RTN | 84.16 | 81.77 | 74.43 | 77.95 | 51.71 | 80.42 | 75.07 |
| | OmniQuant[§] | 83.33 | 81.83 | 73.72 | 79.25 | 52.90 | 80.96 | 75.33 |
| | LSQ (our impl.) | 80.40 | 80.63 | 73.16 | 76.26 | 48.72 | 78.24 | 72.90 |
| | PEQA (our impl.) | 80.89 | 81.72 | 73.80 | 75.42 | 48.46 | 79.76 | 73.34 |
| | **LR-QAT (ours)** | **83.55** | **81.61** | **74.51** | **78.28** | **52.90** | **80.84** | **75.28** |
| W3 pc | RTN | 68.13 | 77.64 | 63.93 | 63.93 | 41.13 | 72.73 | 64.58 |
| | OmniQuant[§] | 74.04 | 78.56 | 65.59 | 70.12 | 42.32 | 73.75 | 67.40 |
| | LSQ (our impl.) | 79.97 | 80.74 | 70.24 | 74.37 | 47.18 | 77.14 | 71.61 |
| | PEQA (our impl.) | 80.03 | 80.09 | 69.93 | 72.90 | 45.82 | 77.32 | 71.02 |
| | **LR-QAT (ours)** | **81.62** | **80.09** | **70.96** | **74.75** | **46.08** | **77.71** | **71.87** |
| W3 g128 | RTN | 78.44 | 79.60 | 69.14 | 71.17 | 43.00 | 74.75 | 69.35 |
| | OmniQuant[§] | 80.31 | 81.39 | 70.48 | 76.85 | 50.51 | 78.82 | 73.06 |
| | LSQ (our impl.) | 81.01 | 80.36 | 71.03 | 74.41 | 47.27 | 77.68 | 71.96 |
| | PEQA (our impl.) | 81.99 | 81.18 | 69.61 | 74.92 | 47.18 | 78.37 | 72.21 |
| | **LR-QAT (ours)** | **81.71** | **80.90** | **70.48** | **75.08** | **47.78** | **78.50** | **72.41** |
| W2 pc | RTN[§] | 39.42 | 51.03 | 49.49 | 26.22 | 26.71 | 26.67 | 36.59 |
| | OmniQuant[§] | 37.80 | 51.63 | 48.30 | 27.78 | 27.22 | 25.59 | 36.39 |
| | PEQA (our impl.)[§] | 74.34 | 75.95 | 66.22 | 62.21 | 38.48 | 69.68 | 64.48 |
| | **LR-QAT (ours)[§]** | **75.08** | **76.39** | **65.98** | **64.48** | **39.59** | **69.14** | **65.11** |
| W2 g128 | RTN[§] | 54.98 | 56.91 | 51.70 | 31.27 | 23.21 | 29.70 | 41.30 |
| | OmniQuant[§] | 57.58 | 67.68 | 55.72 | 42.17 | 27.65 | 51.94 | 50.46 |
| | PEQA (our impl.)[§] | 75.41 | 75.84 | 64.96 | 65.45 | 38.23 | 68.00 | 64.65 |
| | **LR-QAT (ours)[§]** | **75.81** | **77.15** | **64.64** | **63.59** | **39.51** | **70.89** | **65.27** |

Table C5: **LM-eval weight-only quantization results for Mistral 7B**. We report zero-shot accuracy of 6 tasks (higher is better). [§]Uses asymmetric weight quantization.

| Model | # Bits (W-A-KV) | Method | BoolQ | PIQA | Winogrande | ARC-e | ARC-c | HellaSwag | Avg. |
|---|---|---|---|---|---|---|---|---|---|
| LLaMA-1 7B | | FP16 | 75.05 | 79.16 | 70.01 | 72.85 | 44.80 | 76.21 | 69.68 |
| | 4-8-8 | RTN | 71.35 | 76.66 | 66.46 | 66.84 | 41.55 | 72.10 | 65.83 |
| | | SmoothQuant | 71.00 | 76.00 | 66.00 | 67.40 | 42.80 | 67.80 | 65.17 |
| | | LLM-QAT | 74.60 | 77.50 | 67.70 | 70.20 | 45.60 | 73.50 | 68.18 |
| | | PEQA (our impl.) | 74.86 | 78.24 | 70.01 | 70.12 | 42.83 | 75.14 | 68.53 |
| | | **LR-QAT (ours)** | **73.76** | **78.51** | **71.19** | **71.09** | **41.81** | **75.10** | **68.58** |
| | 4-8-4 | RTN | 68.81 | 75.46 | 62.12 | 62.46 | 39.51 | 68.33 | 62.78 |
| | | SmoothQuant | 54.70 | 55.40 | 51.50 | 43.90 | 27.70 | 38.90 | 45.35 |
| | | LLM-QAT | 69.50 | 75.40 | 64.60 | 66.00 | 43.80 | 69.20 | 64.75 |
| | | PEQA (our impl.) | 72.97 | 77.80 | 67.72 | 67.13 | 40.27 | 73.35 | 66.54 |
| | | **LR-QAT (ours)** | **73.64** | **77.91** | **67.56** | **69.28** | **41.30** | **73.25** | **67.16** |
| | 4-4-4 | RTN | 50.49 | 64.25 | 52.41 | 48.27 | 30.12 | 52.04 | 49.60 |
| | | SmoothQuant | 49.10 | 49.80 | 48.00 | 30.40 | 25.80 | 27.40 | 38.42 |
| | | LLM-QAT | 61.30 | 51.50 | 51.90 | 27.90 | 23.90 | 31.10 | 41.27 |
| | | LLM-QAT + SQ | 62.40 | 55.90 | 50.60 | 35.50 | 26.40 | 47.80 | 46.43 |
| | | Outlier Suppression+ | 60.21 | 62.73 | 52.96 | 39.98 | 30.29 | 44.39 | 48.43 |
| | | OmniQuant[§] | 63.51 | 66.15 | 53.43 | 45.20 | 31.14 | 56.44 | 52.65 |
| | | PEQA (our impl.) | 65.69 | 72.31 | 59.83 | 56.52 | 34.22 | 61.79 | 58.39 |
| | | **LR-QAT (ours)** | **67.16** | **71.76** | **59.59** | **58.42** | **34.73** | **62.34** | **59.00** |
| LLaMA-2 7B | | FP16 | 77.74 | 79.11 | 69.14 | 74.58 | 46.25 | 75.98 | 70.47 |
| | 4-8-8 | RTN | 75.87 | 77.91 | 67.88 | 71.09 | 44.03 | 74.51 | 68.55 |
| | | PEQA (our impl.) | 77.37 | 77.97 | 69.77 | 70.54 | 43.52 | 75.50 | 69.11 |
| | | **LR-QAT (ours)** | **77.00** | **78.13** | **69.14** | **72.10** | **44.11** | **75.44** | **69.32** |
| | 4-8-4 | RTN | 70.37 | 76.01 | 63.38 | 68.94 | 41.47 | 70.76 | 65.16 |
| | | PEQA (our impl.) | 74.71 | 77.48 | 67.40 | 69.28 | 42.75 | 73.75 | 67.56 |
| | | **LR-QAT (ours)** | **74.46** | **77.69** | **68.51** | **69.78** | **42.75** | **73.82** | **67.84** |
| | 4-4-4 | RTN | 57.86 | 64.91 | 54.46 | 49.62 | 31.83 | 51.83 | 51.75 |
| | | PEQA (our impl.) | 67.09 | 70.67 | 60.06 | 54.80 | 32.17 | 62.76 | 57.93 |
| | | **LR-QAT (ours)** | **66.94** | **71.98** | **60.77** | **57.20** | **33.87** | **63.10** | **58.98** |
| LLaMA-2 13B | | FP16 | 80.55 | 80.52 | 72.22 | 77.44 | 48.98 | 79.38 | 73.18 |
| | 4-8-8 | RTN | 79.24 | 79.27 | 70.01 | 75.51 | 48.29 | 76.92 | 71.54 |
| | | PEQA (our impl.) | 79.02 | 80.20 | 71.19 | 76.60 | 48.72 | 79.22 | 72.49 |
| | | **LR-QAT (ours)** | **81.10** | **79.76** | **71.35** | **77.36** | **50.60** | **78.91** | **73.18** |
| | 4-8-4 | RTN | 77.25 | 76.61 | 66.69 | 67.72 | 41.13 | 72.98 | 67.06 |
| | | PEQA (our impl.) | 78.01 | 79.22 | 69.30 | 75.59 | 48.21 | 77.78 | 71.35 |
| | | **LR-QAT (ours)** | **78.59** | **79.54** | **70.80** | **75.29** | **48.04** | **77.64** | **71.65** |
| | 4-4-4 | RTN | 62.60 | 67.90 | 53.20 | 57.32 | 34.13 | 55.25 | 55.07 |
| | | PEQA (our impl.) | 68.72 | 74.21 | 62.35 | 63.85 | 36.01 | 68.44 | 62.26 |
| | | **LR-QAT (ours)** | **70.64** | **73.88** | **63.14** | **61.78** | **38.14** | **68.31** | **62.65** |

Table C6: **LM-eval weight and activation quantization results for LLaMA models**. We report zero-shot accuracy of 6 tasks (higher is better). [§]Uses asymmetric weight quantization.

# D ADDITIONAL RESULTS

In this section, we provide extended results and present some additional ablation studies:

- In Table D1, we show a comparison between min-max and the best range setting used for round-to-nearest (RTN) initialization.

- In Table D2, we perform an ablation study comparing different range initializations with RTN (best, as used in the main set of results vs. min-max) for LR-QAT and PEQA applied to LLaMA-2 7B.

- In Table D3, we provide a detailed comparison of weight-only quantization results with BitDistiller (Du et al., 2024).

| # Bits | Range estimator | WikiText-2 perplexity ↓ | | | | | Avg. zero-shot accuracy ↑ | | | | |
|---|---|---|---|---|---|---|---|---|---|---|---|
| | | L1-7B | L2-7B | L2-13B | L3-8B | M-7B | L1-7B | L2-7B | L2-13B | L3-8B | M-7B |
| FP16 | | 5.68 | 5.47 | 4.88 | 6.14 | 5.25 | 69.68 | 70.47 | 73.18 | 74.22 | 75.69 |
| W4 pc | best est. | $L^4$ | $L^{3.5}$ | $L^{3.5}$ | $L^{3.5}$ | $L^4$ | $L^4$ | $L^{3.5}$ | $L^{3.5}$ | $L^{3.5}$ | $L^4$ |
| | best | 6.33 | 6.14 | 5.21 | 7.53 | 5.91 | 68.51 | 68.88 | 71.73 | 72.19 | 73.44 |
| | min-max | 6.85 | 7.14 | 5.40 | 10.53 | 6.33 | 66.23 | 66.41 | 72.19 | 67.44 | 71.84 |
| W4 g128 | best est. | $L^5$ | min-max | min-max | $L^4$ | $L^5$ | $L^5$ | min-max | min-max | $L^4$ | $L^5$ |
| | best | 6.05 | 5.78 | 5.04 | 6.96 | 5.49 | 68.93 | 69.75 | 72.94 | 72.30 | 75.07 |
| | min-max | 6.08 | 5.78 | 5.04 | 6.99 | 5.51 | 68.96 | 69.75 | 72.94 | 72.95 | 74.98 |
| W3 pc | best est. | $L^{3.5}$ | $L^{3.5}$ | $L^5$ | $L^{3.5}$ | $L^4$ | $L^{3.5}$ | $L^{3.5}$ | $L^5$ | $L^{3.5}$ | $L^4$ |
| | best | 12.88 | 26.73 | 8.71 | 34.10 | 9.49 | 54.66 | 43.87 | 55.01 | 47.46 | 64.58 |
| | min-max | 2.4e4 | 1.9e4 | 2.3e3 | 1.6e5 | 3.2e3 | 36.02 | 35.71 | 37.85 | 35.78 | 36.78 |
| W3 g128 | best est. | $L^5$ | $L^4$ | $L^5$ | $L^5$ | $L^5$ | $L^5$ | $L^4$ | $L^5$ | $L^5$ | $L^5$ |
| | best | 7.95 | 7.61 | 6.20 | 15.11 | 6.77 | 63.50 | 63.20 | 67.60 | 57.74 | 69.35 |
| | min-max | 8.10 | 8.22 | 6.14 | 29.38 | 7.22 | 62.69 | 64.07 | 66.81 | 54.54 | 68.35 |
| W2 pc§ | best est. | $L^{2.4}$ | $L^3$ | $L^3$ | $L^{3.5}$ | $L^{3.5}$ | $L^{2.4}$ | $L^3$ | $L^3$ | $L^{3.5}$ | $L^{3.5}$ |
| | best | 4.9e3 | 5.2e3 | 5.2e3 | 6.4e4 | 6.8e3 | 37.92 | 36.52 | 36.27 | 36.80 | 36.59 |
| | min-max | 1.1e5 | 2.5e4 | 4.9e4 | 1.4e6 | 7.5e4 | 39.07 | 36.53 | 39.01 | 37.69 | 38.13 |
| W2 g128§ | best est. | $L^4$ | $L^{3.5}$ | $L^5$ | $L^4$ | $L^5$ | $L^4$ | $L^{3.5}$ | $L^5$ | $L^4$ | $L^5$ |
| | best | 708 | 2.5e3 | 115.6 | 1.4e4 | 369 | 39.74 | 37.94 | 41.12 | 36.97 | 41.30 |
| | min-max | 3.6e3 | 5.9e3 | 341 | 2.8e5 | 3.4e3 | 37.06 | 36.78 | 40.30 | 37.18 | 37.16 |

Table D1: **A comparison between min-max and the best range setting used for round-to-nearest (RTN) initialization for LLaMA and Mistral models**. We report WikiText-2 test set perplexity (lower is better) and average zero-shot accuracy (higher is better). §Uses asymmetric weight quantization.

| # Bits | Method | WikiText-2 perplexity ↓ | | Average zero-shot accuracy ↑ | |
|---|---|---|---|---|---|
| | | best | min-max | best | min-max |
| W4 pc | RTN | 6.14 | 7.14 | 68.88 | 66.41 |
| | PEQA (our impl.) | 5.71 | 5.93 | 69.23 | 68.52 |
| | **LR-QAT (ours)** | **5.66** | **5.85** | **69.72** | **68.75** |
| W4 g128 | RTN | 5.78 | 5.78 | 69.75 | 69.75 |
| | PEQA (our impl.) | 5.67 | 5.67 | 69.64 | 69.64 |
| | **LR-QAT (ours)** | **5.59** | **5.59** | **69.88** | **69.88** |
| W3 pc | RTN | 26.7 | 1.9e3 | 43.87 | 35.71 |
| | PEQA (our impl.) | 6.45 | 11.3 | 65.44 | 55.88 |
| | **LR-QAT (ours)** | **6.13** | **7.99** | **67.66** | **61.69** |
| W3 g128 | RTN | 7.61 | 8.22 | 63.20 | 64.07 |
| | PEQA (our impl.) | 6.05 | 6.41 | 68.10 | 65.88 |
| | **LR-QAT (ours)** | **5.99** | **6.37** | **67.98** | **65.86** |

Table D2: **Ablation study comparing different RTN initialization (best vs. min-max) for LR-QAT and PEQA on LLaMA-2 7B.** We report WikiText-2 test set perplexity (lower is better) and average zero-shot accuracy (higher is better).

| Model | # Bits | Method | PIQA | Winogrande | ARC-c | HellaSwag | Average |
|-------|--------|--------|------|-----------|-------|-----------|---------|
| LLaMA-2 7B | FP16 | | 79.11 | 69.14 | 46.25 | 57.13 | 62.91 |
| | W3 g128 | BitDistiller[§] | 76.99 | 68.35 | 41.21 | 55.38 | 60.48 |
| | | **LR-QAT (ours)** | **77.31** | **68.98** | **42.58** | **55.05** | **60.98** |
| | W2 g128 | BitDistiller[§] | 73.61 | 61.09 | 33.27 | 48.70 | 54.17 |
| | | **LR-QAT (ours)[§]** | **73.83** | **63.69** | **34.98** | **49.72** | **55.56** |
| LLaMA-2 13B | FP16 | | 80.52 | 72.22 | 49.23 | 60.05 | 65.51 |
| | W3 g128 | BitDistiller[§] | 78.67 | 71.59 | 46.67 | 58.66 | 63.90 |
| | | **LR-QAT (ours)** | **79.60** | **70.64** | **46.76** | **58.84** | **63.96** |
| | W2 g128 | BitDistiller[§] | 75.84 | 65.90 | 37.46 | 51.30 | 57.63 |
| | | **LR-QAT (ours)[§]** | **77.86** | **65.98** | **41.04** | **53.86** | **59.69** |

Table D3: **LM-eval weight-only quantization comparison with BitDistiller for LLaMA-2 7B and 13B**. We report zero-shot accuracy of 4 tasks (higher is better). Specifically, we report `acc` for Winogrande, HellaSwag and `acc_norm` for PIQA and ARC-c. [§]Uses asymmetric weight quantization.

# E    INFERENCE SIMULATION RESULTS

In this section, we present results of our inference runtime simulation to measure the overhead incurred when low-rank auxiliary matrices $A$ and $B$ are not fused into the pretrained weight matrix $W_{\mathbb{Z}}$. Additionally, we conduct a preliminary experiment showing that, after the model is trained with QLoRA-style approach, naively fusing high precision $A$ and $B$ into the low-bit pretrained weights $W_{\mathbb{Z}}$ will lead to a significant accuracy drop.

**Inference runtime overhead**    We follow the experimental settings of QLoRA, except using integer quantization as opposed to NF non-uniform quantization format. Starting from a BF16 baseline LLaMA 7B model, we apply `bits-and-bytes`[6] INT8 weight quantization to all linear layers. Note that we don't apply activation quantization and so, even if weights are stored in INT8, we must dequantize them to BF16 before applying matrix multiplication. Finally, similarly to QLoRA, we applied LoRA using the PEFT library[7], with rank $r$ ranging from 0 (i.e., LoRA weights fused) up to rank 32. The results reported in Table E1 show a consistent non-negligible runtime overhead when employing unfused inference, compared to the case when $A$ and $B$ are fused to the original pretrained weights, even with rank $r = 1$.

| Sequence length | LoRA rank, $r$ | Inference runtime, ms | Relative overhead |
|-----------------|----------------|----------------------|-------------------|
| 1024 | 0 (fused) | $154.4^{\pm 3.2}$ | +0% |
| 1024 | 1 | $179.3^{\pm 1.8}$ | +16.1% |
| 1024 | 4 | $183.2^{\pm 3.3}$ | +18.7% |
| 1024 | 32 | $183.0^{\pm 2.1}$ | +18.5% |
| 2048 | 0 (fused) | $229.4^{\pm 0.3}$ | +0% |
| 2048 | 1 | $258.8^{\pm 1.0}$ | +12.3% |
| 2048 | 4 | $260.2^{\pm 1.5}$ | +13.8% |
| 2048 | 32 | $258.1^{\pm 1.2}$ | +12.5% |

Table E1: **Inference runtime simulation for LLaMA 7B with weight-only integer quantization**, using fused and unfused adapters of different ranks. We report the wall time of a single forward pass for a single sequence of specified length on NVidia A100 GPU, averaged over 50 runs. Rank 0 represents the runtime for techniques fusing $A$ and $B$ into the original weights (like LR-QAT).

Note that reported results are targeting a specific hardware (Nvidia A100) and will look different when the model is deployed elsewhere. In case of an actual integer inference hardware, the runtime overhead will likely be even higher, because the speed of $W_{\mathbb{Z}} \cdot \mathbf{x}$ can be greatly decreased by using integer-only multiplications while the overhead of high precision multiplication $AB\mathbf{x}$ will remain

---

[6]https://github.com/bitsandbytes-foundation/bitsandbytes
[7]https://github.com/huggingface/peft

roughly the same. Conversely, our method, which incorporates fused low-rank auxiliary matrices, does not introduce any additional inference latency compared to PTQ, full-model QAT or any other uniform affine quantization methods.

**Naive fusion of QLoRA adapters** We conducted a preliminary experiment to investigate the effect of naively fusing high precision adapters $A$ and $B$ into a low-bit integer weights $W_{\mathbb{Z}}$, after the model has been trained with QLoRA-like technique. For this experiment, we trained a smaller OPT-1.3B (Zhang et al., 2022) model on Wikipedia training set for 1000 steps with effective batch size 32, and report the WikiText-2 validation set perplexity using sequence length 1024. The rest of the hyperparameters are the same as for our main set of experiment. We demonstrate the results of the aforementioned approach and compare it against several baselines, RTN, full-model QAT, and our method in Table E2.

| #Bits | Method | Wikitext-2 Perplexity ↓ |
|---|---|---|
| FP16 | | 29.63 |
| W4 pc | RTN | 37.22 |
| W4 pc | Full-model QAT (LSQ) | 21.66 |
| W4 pc | QLoRA + Naive $A$, $B$ fusion | 26.94 |
| **W4 pc** | **LR-QAT (ours)** | **19.87** |

Table E2: A comparison between naively fusing $A$ and $B$ in QLoRA, our method, RNT, and full-model QAT.

It is easy to see fusing $A$ and $B$ naively after training with QLoRA leads to big increase in perplexity. In contrast, LR-QAT allows for $A$ and $B$ to be fused in the original low-bit pretrained weights seamlessly and without any predictive performance degradation, achieving both an accuracy comparable with full-model QAT and without sacrificing inference performances, as shown in previous paragraph.

# F TRAINING RUNTIME

| # Bits | Method | Per-device batch size × grad. accumulation steps | Time/100 steps, sec |
|---|---|---|---|
| FP16 | Full-model training | $1\times32$ | $974^{\pm3}$ |
| FP16 | LoRA, $r = 32$ | $1\times32$ | $950^{\pm3}$ |
| W4 pc | Full-model QAT | $1\times32$ | $\times$ |
| W4 pc | Full-model QAT + CPU opt. offloading | $1\times32$ | $7426^{\pm216}$ |
| W4 pc | Full-model QAT + CPU opt. offloading | $2\times16$ | $\times$ |
| W4 pc | Full-model QAT + checkpointing | $1\times32$ | $3248^{\pm7}$ |
| W4 pc | Full-model QAT + checkpointing | $2\times16$ | $\times$ |
| W4 pc | LR-QAT, $\varphi = $ Q4.4, $r = 32$ | $1\times32$ | $2938^{\pm6}$ |
| W4 pc | LR-QAT, $\varphi = $ Q4.4, $r = 32$ | $4\times8$ | $1522^{\pm5}$ |
| W4 pc | LR-QAT, $\varphi = $ Q4.4, $r = 1$ | $4\times8$ | $1519^{\pm6}$ |
| W4 pc | LR-QAT, $\varphi = $ Q4.4, $r = 4$ | $4\times8$ | $1528^{\pm3}$ |
| W4 pc | LR-QAT, $\varphi = $ Q4.4, $r = 256$ | $4\times8$ | $1546^{\pm4}$ |
| W4 pc | LR-QAT, $\varphi = $ INT4, $r = 32$ | $4\times8$ | $1518^{\pm8}$ |
| W4 g128 | LR-QAT, $\varphi = $ Q4.4, $r = 32$ | $4\times8$ | $1528^{\pm5}$ |

Table F1: **Training runtime comparison** between LR-QAT, full-model QAT (LSQ), full-precision training and full-precision LoRA for LLaMA 7B on Nvidia A100 80GB GPU, assuming effective batch size 32 and sequence length 1024. We repeat each experiment 5 times and report mean ± standard deviation. $\times$ denotes out of memory.

| Method | GPU mem., GB | Runtime, hours | WikiText-2 ppl. ↓ | | Zero-shot acc. ↑ | |
|---|---|---|---|---|---|---|
| | | | W4 pc | W3 pc | W4 pc | W3 pc |
| Full-model QAT (LSQ), $10^4$ iters | 62.2 (98.5) | $90.2^{\pm0.2}$ | $5.77^{\pm0.02}$ | $6.14^{\pm0.01}$ | $68.96^{\pm0.29}$ | $67.14^{\pm0.13}$ |
| PEQA, $10^4$ iters | 19.9 | $35.6^{\pm0.22}$ | 5.71 | 6.45 | 69.23 | 65.44 |
| OmniQuant[§] | **11.6** | **1.21** | 5.74 | 6.58 | 68.19 | 63.94 |
| **LR-QAT (ours)**, $10^4$ iters | 20.5 | $42.3^{\pm0.13}$ | **$5.66^{\pm0.00}$** | **$6.13^{\pm0.02}$** | **$69.72^{\pm0.32}$** | **$67.70^{\pm0.25}$** |
| **LR-QAT (ours)**, $10^3$ iters | 20.5 | 4.23 | 5.68 | 6.22 | 69.43 | 66.93 |

Table F2: A comparison of the proposed method ($\varphi = $ Q4.4) with the full-model QAT, PEQA and OmniQuant on LLaMA-2 7B with W4 and W3 per-channel quantization. [§]Uses asymmetric weight quantization. We report mean and standard deviation over 5 runs with different random seeds. We also report the maximum GPU memory with (without) gradient checkpointing and the training runtime on a Nvidia A100 80GB GPU.

