# OpenReview forum: "Low-Rank Quantization-Aware Training for LLMs"
_ICLR.cc/2025/Conference — Submitted to ICLR 2025_

### Official Review · Reviewer_bMgZ · 2024-10-28

**Soundness:** 3
**Presentation:** 3
**Contribution:** 2
**Rating:** 5
**Confidence:** 5

**Summary:**

The paper proposes Low-Rank Quantization-Aware Training (LR-QAT) to address the memory and computational inefficiencies of traditional quantization-aware training (QAT) methods when applied to large language models (LLMs). LR-QAT integrates low-rank adapters into the quantization process and utilizes fixed-point representations and gradient checkpointing to minimize memory usage. Applied to several LLM architectures, LR-QAT achieves model performance on par with full-model QAT, while being resource-efficient and inference-friendly, making it feasible for deployment on consumer-grade GPUs.

**Strengths:**

1- **Efficiency in Memory and Computation**: The approach effectively reduces the memory requirements for QAT, enabling large models to be trained on a single 24GB GPU without compromising predictive performance.

2- **Inference Efficiency**: Unlike many related low-rank and quantization methods, LR-QAT allows the fused low-bit pretrained model to be used directly in inference without additional overhead, which is a practical advantage.

3- **Comprehensive Experimental Validation**: The authors evaluate LR-QAT on multiple LLM families and bitwidth settings, demonstrating versatility across different quantization and low-rank configurations.

4- **Analytical Justifications**: The paper offers theoretical reasoning for observed outcomes, such as the choice of downcasting operators and low-rank approximation within the quantization operation, which contributes to understanding the method's impact on LLM quantization.

**Weaknesses:**

1- **Limited Comparison in Initialization**: The paper relies on round-to-nearest (RTN) for initializing quantized weights without extensive exploration of alternatives. AdaRound [1] has shown that RTN is not necessarily and optimal quantization scheme, and even searching over random quantization schemes can lead to better results. The authors even pointed our using other method such as OmniQuant as a possible initialization scheme.

2- **Lack of Complete Results in Tables**: In Tables 4 and 5, several cells contain dashes (“-”), implying missing data. This restricts fair comparison across methods and limits insight into LR-QAT’s performance in certain configurations.

3- **Potential for Limited Practical Impact**: In cases where LR-QAT's fine-tuning time is significant, the marginal performance gain over zero-shot methods may not justify the overhead, questioning the scalability and applicability of LR-QAT in high-stakes deployments. A comparison between the quantization time required for different methods can help shed light to the trade off between different methods (in addition to ther fine-tuning timing results reported in the appendix).

4- **Fixed Point Representation Modeling**: I believe that the fixed point representation of the weights can be modeled using the integer representation with tuning of scaling parameters. This could potentially yield better accuracy and flexibility without increasing the complexity of implementation.

**Typos and Minor Issues**
- Please ensure consistent notation for mathematical symbols throughout equations.
- Avoid using “paragraph” as a placeholder, as seen in Section 2.

[1] Nagel et al., "Up or Down? Adaptive Rounding for Post-Training Quantization," ICML 2020

**Questions:**

1- **Choice of $s_0$ in Equation 5**:  Could the authors clarify how  $s_0$ is chosen? Is it based on a range estimation technique or empirical tuning?

2- **Scaling Parameters for Integer Representation**:  I believe that the fixed point representation of the weights can be modeled using the integer representation with tuning of scaling parameters. Have the authors experimented with tuning the scaling parameters?

3- **Ignoring Quantization in Backward Pass**: Would omitting quantization for weights and low-rank adapters during the backward pass yield notable memory or computation savings?

4- **Trade-offs with Gradient Checkpointing**: I'm wondering what the effects of ignoring quantization of the weights (and low-rank adapters) are in the backward pass. This could lead to both memory and compute savings in the backward pass, since it allows efficient computation of low-rank multiplications.

5- **Adapter Rank and Dataset Size Scaling**: Based on Figure 2, the perplexity of the model w.r.t. the adapter rank is almost flat, and has perturbations in higher ranks. As the authors have pointed out, that might be an issue with the size of the fine-tuning set. I suggest the authors use scaling laws for finding the dataset sizes for different low-rank adapters and check if that solves the inconsistency with these results.

6- **Alternative Initializations for Quantized Weights**: Has the paper considered initialization methods beyond RTN? OmniQuant could offer improved results by leveraging adaptive rounding and error minimization techniques, respectively.

---

> ### Author Response · Authors · 2024-11-20
> **General response**
>
> We thank the reviewer for their thoughtful and useful feedback. Please, find our replies to specific concerns below.
>
> ## W1: Limited Comparison in Initialization
>
> We agree that RTN is not necessarily an optimal quantization scheme, though it is the most common initialization choice in QAT literature.
>
> Please note that our RTN baseline is significantly stronger than the ones from related work (e.g., OmniQuant, LLM-QAT), because we set the ranges based on minimizing the $L^p$-norms between quantized and unquantized weights and use the best performing $L^p$-norm based on a holdout Wikipedia validation set (see the details in L819-823 and Table C8).
>
> We did perform an additional ablation study comparing different range initializations with RTN for LR-QAT and PEQA applied to LLaMA-2 7B (best, as reported in the paper vs. using min-max initialization), which we included in the subsequent comment.
>
> Note that it is rather uncommon to use some of the learnable rounding techniques (such as GPTQ, AdaRound) as initialization for the end-to-end QAT setting. At the same time, we see the value in exploring the combination of our method with techniques like OmniQuant as a preprocessing step. We will further extend our ablation study to include OmniQuant for the next revision.
>
> ## W2: Lack of Complete Results in Tables
>
> We agree that having more than a few missing results in the tables makes it more difficult to compare results across methods and limits the insight into performance in certain configurations.
>
> Please note that we put a significant effort in reimplementing and reporting results for LSQ and PEQA, which we deem the most relevant baselines given that they are also QAT-based techniques. We also made sure to account for initialization, number of training examples, hyperparameter tuning etc., to ensure fairness in this comparison. In addition to that, we report results across 5 models, 6 distinct weight-only and 3 weight and activation quantization settings and compare against numerous baselines, which is substantially more than in many related works.
>
> Note that a good fraction of missing results corresponds to LLaMA-3 and Mistral model families, for which there were no reported results in most related work. In addition to that, unfortunately, some prior works were not consistent in reporting results across all model families and settings (e.g., OmniQuant reports WikiText-2 perplexity but not zero-shot accuracy for W-only results and vice versa for W4A4 setting).
>
> While we agree that authors should put in a reasonable effort to compare fairly to related work, which may include to reimplement and/or run some new studies with existing methods, we do not think that it is reasonable to expect from authors to always reimplement or rerun all relevant baselines for all settings. We do not think that it is reasonable to expect from authors of new papers to always reimplement and/or run all the missing baselines.
>
> Having said that, we agree to generate and include the missing results from OmniQuant, as it is the strongest PTQ baseline that consistently outperforms other PTQ techniques (GPTQ, AWQ, RTN). Please see the below comment for additional zero-shot accuracy results for OmniQuant for LLaMA-1 and LLaMA-2 model families.
>
> ## W3: Potential for Limited Practical Impact
>
> Please see general response to all reviewers. In short, even with just ~4h runtime (1k steps) we outperform OmniQuant, which for many practical cases we believe is valuable trade-off.
>
> ## W4: Fixed Point Representation Modeling
>
> This is an interesting observation and suggestion.
>
> Indeed, this is exactly how fixed-point numbers, such as Q4.4, are defined and how we implemented them: the set of numbers representable by Q4.4 is the same as the set of numbers representable by INT8, divided by $2^4$ (or more generally, divided by $2^{8-b}$, where $b$ is the target bitwidth).
>
> Hence $\Phi_0$ can be written as $2^{-(8-b)}\cdot W_{\text{INT8}}$, where $W_{\text{INT8}}$ is the frozen INT8 matrix.
> Note, $W_{\text{INT8}}$ needs to be frozen in order to get the memory savings, thus the scale used to obtain $W_{\text{INT8}}$ cannot be tuned.
>
> However, by tuning the scaling factor $\tau=2^{-(8-b)}$ but still keeping the $W_{\text{INT8}}$ frozen, we could in principle achieve slightly better accuracy, however, the term $\tau \cdot W_{\text{INT8}}$ would lose its interpretation of being an approximation to the original weights with the fractional part being truncated to $8-b$ bits.
>
> ## typos and minor issues
>
> All fixed in the updated version, thanks!

---

> ### Author Response · Authors · 2024-11-20
> **Clarifications for the questions**
>
> ## Q1
>
> $s_0$ is the frozen value of the scale at initialization (i.e., right after the range estimation).
>
> ## Q2
>
> Please see response for W4 regarding the relationship between fixed-point and integer representation.
>
> ## Q3
>
> To make sure we understand your question correctly, do you mean that during the backward pass we would pretend that $s \cdot \text{clip}(\text{round}(\Phi_0 + AB),\ldots)$ is $s \cdot (\Phi_0+AB)$ , i.e. ignore both clipping and rounding?
>
> If so,
> * in terms of memory, no, as we are already making sure that we are not storing intermediate results of quantization using gradient checkpointing (see Figure 1).
> * in terms of computation, there might be some minor savings when computing gradients w.r.t $A$, $B$ since now we will be ignoring not only rounding (by using STE assumption, L218) but also clipping. On the other hand, this could potentially lead to less accurate gradients depending on the range of $\Phi_0+AB$.
>
> ## Q4
>
> Please see response to Q3.
>
> ## Q5
>
> Please note that the size of the fine-tuning dataset itself is not the issue, as we are using a general pre-training dataset, but rather the number of iterations we used for this study.
>
> As we focus on efficient QAT techniques, we did not explore significantly longer fine-tuning. Indeed, if we would consider significantly longer fine-tuning, then applying scaling laws would be a very interesting direction.
>
> ## Q6
>
> Please see the response to W1.

---

> > ### Comment · Reviewer_bMgZ · 2024-11-27
> >
> > Dear Authors,
> >
> > Thank you for your thoughtful responses to my questions and concerns. I appreciate the detailed discussion and look forward to seeing these clarifications and improvements incorporated into a revised version of your manuscript, where I can analyze all the results altogether. I believe that for proper comparison of different methods, sufficient data should be provided.
> >
> > I wanted to clarify three point about my questions:
> >
> > **Fixed Point Representation Modeling:** We both agree that the fixed-point representaion of the numbers can be modeled using the integer representation with tuning (not necessarily learning it by during fine-tuning) of scaling parameters. My main concern is why you are limiting your scale to positive or negative powers of 2, while the scaling factor can be any real number that can be tuned for better accuracy.
> >
> > **Clarification about Q3:** Yes, your assumption about my understanding is correct. By doing that, you won't need to compute the quantization values with checkpointing again, which can help reduce a lot of computations for generating large low-rank matrices.
> >
> > **Clarification about Dataset Size:** I apologize for the confusion. I'm not concerned about the original size of the datasets, but I'm about the number of tokens used for fine-tuning the models. And by the *datset size* I was refering to the number of tokens you have used for fine-tuning. Thank you for clarifying that you haven't tested different dataset sizes.

---

> > > ### Author Response · Authors · 2024-11-28
> > >
> > > Dear Reviewer,
> > >
> > > We are glad to address your questions and concerns and sincerely appreciate your thoughtful feedback and further clarifications.
> > >
> > > Please, refer to the latest general comment on the updated revision of our manuscript, where we tried to include as many recommendations as possible (& promise to include the rest in the final revision).
> > >
> > > **Fixed Point Representation Modeling**: Indeed, we agree that this is an additional degree of freedom that is interesting to exploit. We do hypothesis that the effect of this might be limited as this can only be done before the start of training (fine-tuning). Nonetheless it is an interesting avenue to explore in future work.
> > >
> > > **Clarification about Q3**: Indeed, we agree that this can save the compute of generating the product of low-rank matrices during checkpointing. While such an approach could reduce some of the checkpointing overhead, as pointed out before, this can potentially lead to less accurate gradients. We are intrigued by this suggestion and consider it as future work to even further improve the compute- and memory-efficiency of our approach.
> > >
> > > Thank you for your constructive suggestions. We hope these updates address your concerns effectively and look forward to your further feedback.
> > >
> > > Best, authors

---

> ### Author Response · Authors · 2024-11-20
> **RTN initialization study**
>
> | # Bits | Method | WikiText-2 (**best RTN**) | WikiText-2 (min-max) | Zero-shot (**best RTN**) | Zero-shot (min-max) |
> |-|-|:-:|:-:|:-:|:-:|
> | FP | - | (5.47) | - | (70.47) | - |
> | W4 pc   | RTN              | **6.14**  | 7.14  | **68.88** | 66.41 |
> | W4 pc   | PEQA (our impl.) | **5.71**  | 5.93  | **69.23** | 68.52 |
> | W4 pc   | LR-QAT           | **5.66**  | 5.85  | **69.72** | 68.75 |
> | W4 g128 | RTN              | **5.78**  | 5.78  | **69.75** | 69.75 |
> | W4 g128 | PEQA (our impl.) | **5.67**  | 5.67  | **69.64** | 69.64 |
> | W4 g128 | LR-QAT           | **5.59**  | 5.59  | **69.88** | 69.88 |
> | W3 pc   | RTN              | **26.7** | 1.9e3 | **43.87** | 35.71 |
> | W3 pc   | PEQA (our impl.) | **6.45**  |  11.3 | **65.44** | 55.88 |
> | W3 pc   | LR-QAT           | **6.13**  |  7.99 | **67.66** | 61.69 |
> | W3 g128 | RTN              | **7.61**  | 8.22  | **63.20** | 64.07 |
> | W3 g128 | PEQA (our impl.) | **6.05**  | 6.41  | **68.10** | 65.88 |
> | W3 g128 | LR-QAT           | **5.99**  | 6.37  | **67.98** | 65.86 |
>
> We report WikiText-2 perplexity (lower is better) and Zero-shot accuracy over 6 tasks (higher is better).
> Note that as listed in Table C8 in the appendix, we found the best range estimators to be L^3.5, min-max, L^4, L^3.5 for w4pc, w4g128, w3pc, w3g128, respectively.
> We will also include this ablation study in the paper.

---

> ### Author Response · Authors · 2024-11-20
> **Extra OmniQuant baseline results**
>
> Below you can find OmniQuant results for LLaMA-1 and LLaMA-2 model families, which we obtained using their public open-sourced code base (https://github.com/OpenGVLab/OmniQuant).
>
> We used the provided checkpoints by the authors and ensured that the evaluation uses exactly the same lm_eval version (0.4.2). With this, we matched the WikiText-2 perplexity numbers reported by OmniQuant closely as well as the FP16 zero-shot baseline reported in our paper.
>
> |	# Bits	|	Method	|	L1-7B	|	L2-7B	|	L2-13B	|
> |-|-|:-:|:-:|:-:|
> |	FP	|		                  |	69.68	|	70.47	|	73.18	|
> |	W4 pc	|	RTN	                  |	68.51	|	68.88	|	71.73	|
> |	W4 pc	|	OmniQuant*	          |	68.48	|	68.19	|	71.69	|
> |	W4 pc	|	**LR-QAT (ours)**	    |	**68.54**	|	**69.72**	|	**73.19**	|
> |	W4 g128	|	RTN	                |	68.93	|	69.75	|	**72.94**	|
> |	W4 g128	|	OmniQuant*	        |	**69.15**	|	69.58	|	72.80	|
> |	W4 g128	|	**LR-QAT (ours)**	  |	**69.15**	|	**69.88**	|	**72.91**	|
> |	W3 pc	|	RTN	                  |	54.66	|	43.87	|	55.01	|
> |	W3 pc	|	OmniQuant*	          |	66.40	|	63.94	|	70.20	|
> |	W3 pc	|	**LR-QAT (ours)**	    |	**66.60**	|	**67.70**	|	**71.22**	|
> |	W3 g128	|	RTN	                |	63.50	|	63.20	|	67.60	|
> |	W3 g128	|	OmniQuant*	        |	66.77	|	67.52	|	70.97	|
> |	W3 g128	|	**LR-QAT (ours)**	  |	**66.81**	|	**68.62**	|	**71.51**	|
> |	W2 pc	|	RTN*	                |	37.92	|	36.52	|	36.27	|
> |	W2 pc	|	OmniQuant*	          |	49.78	|	43.67	|	49.72	|
> |	W2 pc	|	**LR-QAT\* (ours)**	  |	**61.77**	|	**60.03**	|	**65.28**	|
> |	W2 g128	|	RTN*	              |	39.74	|	37.94	|	41.12	|
> |	W2 g128	|	OmniQuant*	        |	54.31	|	52.00	|	57.16	|
> |	W2 g128	|	**LR-QAT\* (ours)**	|	**61.60**	|	**61.70**	|	**66.75**	|
>
> We report zero-shot accuracy over 6 tasks (higher is better). \*Uses asymmetric quantization.
>
> As we can see, LR-QAT is consistently outperforms OmniQuant across all models and settings (except W4g128 in which they are on par) and the gap becomes progressively bigger as we approach more extreme 2-bit quantization.
>
> We will include these results in Table 4 in the updated revision of the paper. In addition to that, we will also generate and include the remaining missing OmniQuant results for LLaMA-3 & Mistral, and for weight-activation quantization and include them in Tables 4 & 5, respectively.

---

### Official Review · Reviewer_NFjd · 2024-10-29

**Soundness:** 2
**Presentation:** 3
**Contribution:** 2
**Rating:** 5
**Confidence:** 4

**Summary:**

Inspired by parameter efficient fine-tuning literature, this paper proposes LR-QAT – a lightweight and memory-efficient QAT algorithm for LLMs. LR-QAT employs several components to save memory without sacrificing predictive performance: (a) low-rank quantization-aware reparameterization; (b) downcasting operation using fixed-point or double-packing and (c) checkpointing.

**Strengths:**

LR-QAT introduces and combines several innovations designed to reduce memory use without sacrificing model performance: (1) a form of QAT with low-rank reparameterization, in which it places the low-rank weights in the integer domain to ensure they align with the quantization grid of the pretrained weights. This allows for seamless fusion during inference into a single low-bit integer matrix. (2) A downcasting operator that represents the frozen pretrained weights as low-bit INT-b (b ≤ 4) double-packed into INT8 or as fixed-point values stored in INT8.

It applies LR-QAT to LLaMA-1/2/3 and Mistral model families and demonstrate its effectiveness on several general language modeling datasets and zero-shot evaluation on some of the common reasoning downstream tasks. The method outperforms recent LLM quantization approaches and reaches the same predictive performance as full-model QAT at the fraction of its memory usage.

**Weaknesses:**

The novelty may be limited. The proposed method combines traditional quantization and LoRA methods. The downcasting operator cast the input to one of pre-existing floating-point formats which follows previous works such as  (Oberstar, 2007) and  (Li et al., 2023).  The gradient checkpointing mainly follow the previous work (Chen et al., 2016). The technical contribution may be limited.

It only has results which finetune on a single dataset. It is better to demonstrate the performance by finetuning on different datasets. For example, QLoRA and QALoRA finetunes on two different datasets. Finetuning on multiple different datasets can demonstrate the performance in general cases. If finetuning on only one single dataset, we are not sure about the general performance of the proposed method on other datasets.

There are some typos such as  'leading to a subpar performance. paragraph' in Line 180.

The claim of 'an extended pretraining method' for this work does not seem to be solid. It seems that 'task-specific fine-tuning' or 'extended pretraining' just depends on which dataset is used for further training. Using downstream datasets is  task-specific fine-tuning    and using pretraining dataset is the so called extended pretraining.  This difference is minor and the proposed method or the baselines can switch the finetuning datasets.  Thus these methods can either be task-specific fine-tuning or extended pretraining. It does not seem to be very difficult to switch the datasets.  It is better to provide more discussion about the claim of extended pretraining.

The method is very similar to QA-LoRA. In QA-LoRA, weights merged with LoRA are still in a quantized form so that LLMs can be deployed with computational efficiency. The core idea of has been investigated in QA-LoRA. It is better to have a more detailed comparison with QA-LoRA to discuss the difference. It is better to compare with QA-LoRA in experiments.

**Questions:**

See the weakness.

---

> ### Author Response · Authors · 2024-11-20
>
> We thank the reviewer for their thoughtful and useful feedback. Please, find our replies to specific concerns below.
>
> ## W1: novelty
>
> Please, find our reply to your concerns regarding the novelty in the general response to all reviewers.
>
> ## W2: finetuning on different datasets
>
> Please note that one of the core ideas of the proposed approach is that it’s positioned as a general extended pretraining method, as opposed to being strictly a fine-tuning on the downstream task method. The resulting model is a low-bit general pretrained LLM backbone, that can still be utilized for any task afterwards or further combined with LoRA-based techniques or other approaches.
>
> At the same time, we agree that demonstrating the performance by fine-tuning on different datasets will further strengthen the experimental section and enable comparisons to more related work, including the aforementioned QLoRA and QA-LoRA.
>
> We will include results of LR-QAT fine-tuned on Alpaca / FLANv2 datasets in the updated version of the paper.
>
> ## W3: typos
>
> All fixed in the updated version, thanks!
>
> ## W4: extended pre-training vs fine-tuning
>
> We agree that the extended pre-training vs task-specific fine-tuning deserves more discussion.
> We chose to ‘position the paper’ as extended pre-training for our motivation and in order to compare fairly to a wide variety of PTQ and QAT literature.
>
> However, as the reviewer correctly points out, our proposed technical solution (LR-QAT) is not limited to this setup which we deem an additional benefit of it. As mentioned above, we will extend the experimental section to include LR-QAT trained on Alpaca / FLAN-v2 to allow comparison to PEFT literature, including QLoRA and QA-LoRA.
>
> ## W5: comparison with QA-LoRA
>
> There are indeed some similarities between QA-LoRA and our work, namely:
> * Both use low-rank (re)parameterization
> * Both target memory- and inference-efficient training
>
> However, there are some significant differences:
> * Key difference, restrictions on quantization granularity: QA-LoRA assumes all rows of A must be the same => weight quantization has to be done with group-wise granularity with very small group-size (e.g., 32) otherwise it’s very inflexible, whereas our method works for any quantization granularity out of the box. In particular, QA-LoRA cannot be applied to pre-channel weight quantization, which is a common case for weights.
> * Because of the aforementioned restrictions, QA-LoRA also reports results only using asymmetric weight quantization, which of course provides extra degree of flexibility. Note that not all hardware can support asymmetric weight quantization, and even if it does, it introduces extra inference overhead compared to symmetric weight quantization [1]. At the same time, we demonstrate state-of-the-art results for W4 and W3 uniform affine quantization using symmetric scheme.
> * The above two points further lead to increased memory usage and compute overhead during inference.
> * Finally, note that QA-LoRA is introduced as a downstream task fine-tuning method (like QLoRA). As a consequence of that, this means they do not compare extensively to other PTQ and QAT techniques.
>
> ## References
> [1] Nagel et al., "A white paper on neural network quantization", ArXiV:2106.08295

---

> > ### Comment · Reviewer_NFjd · 2024-12-02
> >
> > Thanks for the rebuttal. After I went through the rebuttal, I still have some concerns.
> >
> > I still believe that it is not very solid to position the paper as extended pre-training. The so-called 'extended pre-training' just needs to switch the dataset and there is no new techniques. The response in the general rebuttal does not highlight the unique contributions of this work.
> >
> > Furthermore, I checked the baselines results in the paper, such as OmniQuant. The results of OmniQuant in this paper are exactly the same as the original OmniQuant paper. The authors generally copy the results of the original OmniQuant as the baselines. But the original OmniQuant adopts WikiText2 as the finetuning dataset. This paper adopts a part of SlimPajama as the finetuning set. The finetuning datasets are different. It may not be appropriate to compare the two methods. We are not sure whether the performance difference is introduced by the different datasets or others. It may be better to discuss this issue.

---

> ### Author Response · Authors · 2024-12-04
>
> Dear Reviewer,
> We thank you for carefully considering our rebuttal and for your valuable feedback. Please, see further clarification below.
>
> **Extended pre-training**: Of course, we fully agree that from a technical perspective, the difference between extended pre-training and fine-tuning is in practice just switching the dataset (and there is no novelty in that).
>
> We would like to clarify the framing of our method as ‘extended pre-training’ technique. As mentioned, our main motivation is to position our method as a general QAT technique and to compare fairly to a wide variety of PTQ and QAT literature. In contrast to PEFT literature, where it’s prevalent to combine the method together with the downstream task fine-tuning, most PTQ and QAT literature does not do any task fine-tuning (all PTQ literature but also LLM-QAT, LSQ etc). Therefore, we would like to very clearly separate and isolate improvements coming from the novelty of the method from the “pipeline improvements” (which includes a use of different, potentially more advantageous dataset, as well as other things like extra loss terms/knowledge distillation etc.).
>
> As mentioned in our earlier response, we will extend the experimental section to include LR-QAT trained on different datasets (such as Alpaca / FLAN-v2) to allow comparison to PEFT literature, including QLoRA and QA-LoRA.
> Following your suggestions and this discussion, we will carefully review and adjust the phrasing in the paper to better reflect our intention and the relationship between fine-tuning and extended pre-training.
>
> **Novelty**: To clarify, the ‘extended pre-training' is not novelty and rather a framing of our work. Though there are clear technical novelties about our approach, including introducing a low-rank reparameterization for QAT that does not lead to any inference time overhead, and the use of a fixed-point downcasting operator, and the use of checkpointing to further decrease memory during training.
> As mentioned in the general response, we agree that our method is relatively straightforward from an implementation standpoint. However, it is a novel QAT technique that is very effective, easy to use, and is solving the important problem of memory- and inference-efficient LLM quantization. Despite its simplicity, we think it is a novel and valuable contribution to the field.
>
> **OmniQuant dataset**: Indeed, our OmniQuant numbers are directly taken from their paper as is a common practice in most literature. We do agree that there might be a small difference between fine-tuning on Wikitext-2 vs fine-tuning on SlimPajama.
>
> Since we introduce a new QAT technique, we deem PEQA and full-model QAT (LSQ) as the most relevant baselines. Please note that to ensure fairness, we carefully followed the same experimental setup (including dataset, but also initialization, number of training examples, hyperparameter tuning etc.) for both PEQA and full-model QAT (LSQ). As we included PTQ techniques mainly as additional references, we did not follow the same practice there and used the authors provided results for comparisons.
>
> Given that OmniQuant is our strongest PTQ baseline, we agree that it is valuable to include extra results using the same calibration dataset – SlimPajama, to ensure fairness.
> Below (in the next comment) you can find OmniQuant results for LLaMA-1/2 model families using SlimPajama (SP) as a calibration set, which we obtained using their public open-sourced code base (https://github.com/OpenGVLab/OmniQuant). We kept the same hyperparameters (number of iterations etc.) as for the original OmniQuant results.
>
> As we can see, WikiText-2 (WT2) perplexity is consistently worse when using SlimPajama as a calibration set (as expected) while zero-shot accuracy stays comparable.
> We will update the paper with these results and include this detailed comparison in the appendix in the updated version of the paper.
>
> We greatly appreciate your time and consideration and hope these updates address your concerns.
>
> Best, authors

---

> ### Author Response · Authors · 2024-12-04
> **OmniQuant results trained on SlimPajama**
>
> |	Model	|	\# Bits	|	WT2 ppl. (D=WT2)	|	WT2 ppl. (D=SP)	|	Δ ppl.	|	Zero-shot (D=WT2)	|	Zero-shot (D=SP)	|	Δ Zero-shot	|
> |-|-|:-:|:-:|:-:|:-:|:-:|:-:|
> |	LLaMA-1 7B	|	W4 pc	|	5.86	|	5.95	|	0.09	|	68.48	|	68.40	|	-0.08	|
> |	LLaMA-1 7B	|	W4 g128	|	5.77	|	5.83	|	0.06	|	69.15	|	69.14	|	-0.02	|
> |	LLaMA-1 7B	|	W3 pc	|	6.49	|	6.83	|	0.34	|	66.40	|	64.69	|	-1.71	|
> |	LLaMA-1 7B	|	W3 g128	|	6.15	|	6.35	|	0.20	|	66.77	|	66.27	|	-0.50	|
> |	LLaMA-1 7B	|	W2 pc	|	15.47	|	25.77	|	10.30	|	49.78	|	48.46	|	-1.31	|
> |	LLaMA-1 7B	|	W2 g128	|	9.72	|	11.66	|	1.94	|	54.31	|	54.58	|	0.28	|
> |	LLaMA-2 7B	|	W4 pc	|	5.74	|	5.93	|	0.19	|	68.19	|	67.74	|	-0.45	|
> |	LLaMA-2 7B	|	W4 g128	|	5.58	|	5.66	|	0.08	|	69.58	|	69.57	|	-0.01	|
> |	LLaMA-2 7B	|	W3 pc	|	6.58	|	7.01	|	0.43	|	63.94	|	64.17	|	0.22	|
> |	LLaMA-2 7B	|	W3 g128	|	6.03	|	6.25	|	0.22	|	67.52	|	67.34	|	-0.18	|
> |	LLaMA-2 7B	|	W2 pc	|	37.37	|	73.49	|	36.12	|	43.67	|	40.57	|	-3.11	|
> |	LLaMA-2 7B	|	W2 g128	|	11.06	|	14.45	|	3.39	|	52.00	|	49.89	|	-2.11	|
> |	LLaMA-2 13B	|	W4 pc	|	5.02	|	5.09	|	0.07	|	71.69	|	72.23	|	0.54	|
> |	LLaMA-2 13B	|	W4 g128	|	4.95	|	5.01	|	0.06	|	72.80	|	72.82	|	0.02	|
> |	LLaMA-2 13B	|	W3 pc	|	5.58	|	5.79	|	0.21	|	70.20	|	69.38	|	-0.81	|
> |	LLaMA-2 13B	|	W3 g128	|	5.28	|	5.45	|	0.17	|	70.97	|	70.93	|	-0.05	|
> |	LLaMA-2 13B	|	W2 pc	|	17.21	|	24.49	|	7.28	|	49.72	|	48.63	|	-1.09	|
> |	LLaMA-2 13B	|	W2 g128	|	8.26	|	9.55	|	1.29	|	57.16	|	57.17	|	0.01	|

---

### Official Review · Reviewer_ipQw · 2024-11-05

**Soundness:** 3
**Presentation:** 3
**Contribution:** 3
**Rating:** 8
**Confidence:** 4

**Summary:**

The authors propose a Low-Rank Quantization Aware Training (LR-QAT) algorithm to train LLM model on single consumer grade GPU. In particular, LR-QAT is based on quantizing low-rank adapters (LoRA) such that it could be fused into quantized frozen pre-trained weight. In addition, it uses a downcasting operator for frozen pretrained weight and gradient checkpointing for tackling increased memory usage owing to product computation of low rank adapters. In contrast to other works on Post training quantization, Quantization-aware training of full models, and PEFT-based (LoRa) quantization schemes, LR-QAT is geared towards better accuracy, memory-efficiency and inference-efficiency (via fusion of low-rank adapter to low-bit integer frozen weight)

**Strengths:**

1. The paper is well-written with adequate background and related work.

2. The paper combines LoRA + QAT techniques and applies down casting and gradient checkpointing for memory-efficient and inference-efficient LLMs

3. The key innovation lies in seamless fusion of the quantized low-rank adapters with the quantization field of frozen retrained weight unlike other LoRA inspired quantization works.

4. The empirical analysis is extensive with sufficient comparisons and ablation studies with impact of rank, and down casting operation.

**Weaknesses:**

1. Figure 1 probably is not referenced in the main text. Probably a deeper discussion on Fig. 1 (right) is needed with respect to goal of training LLMS on single device with 24GB.

2. It might be helpful to have a comparison table or figure that depicts the what parts of the model are being quantized and quantization scheme used across various PTQ, QAT and LoRa-inspired works referenced and compared in experiments - this would help put related works in perspective.

3. LoRA is predominantly used for fine-tuning as part of PEFT methods, but in many places, the authors use training, pertaining and fine-tuning with LoRA interchangeably. It would be helpful to be consistent with keywords related to LoRA and proposed LR-QAT. eg. Line 187, should it be fine-tuning compared to training, title of the work, quantization-aware training or pretraining of LLMs when authors claim LR-QAT to be general purpose extended pretraining method versus fine-tuning method

4. PTQ and LR- abbreviations used in abstract without full form.

5. Line 48: Please specify which is most related work the authors are contrasting with, since they referenced both PEQA (Line 176) and QA-LoRA (Line 195) to be closest to theirs.

6. Where does PEQA fit in Table 1?

7. Lines 049 and 177: Define inference efficiency of QAT here earlier than Lines 201-202

8. Line 65: Define r as "rank"

9. Line 193: Typo "is" ..This issue.....

**Questions:**

The questions have been shared in earlier section"Weakness"

---

> ### Author Response · Authors · 2024-11-20
>
> We thank the reviewer for their thoughtful and useful feedback. Please, find our replies to specific concerns below.
>
> ## W1: referencing Figure 1
>
> That is a very good catch! Indeed, we did not reference Figure 1, and we will make sure to include it in the updated version of the paper.
>
> We agree that a more detailed discussion of the memory breakdown (Figure 1, right) is valuable to the paper as one of our primary goals is to enable memory efficient QAT. We will extend section 5.3 (comparison to QAT) with an additional paragraph discussing this.
>
> ## W2: a table listing quantization settings
>
> Note that we follow closely prior literature, specifically OmniQuant and LLM-QAT, in our quantization settings and assumptions. In case the two differed (only one activation quantizer in self-attention), we used a superset of both, thus our numbers are comparable in a fair manner to both works.
>
> We do agree that including a comparison table that lists what parts of the model are being quantized together with the rest of quantization settings and assumptions across reported methods will be valuable and will provide a more complete picture. We will include this in the appendix in the updated version of the paper.
>
> ## W3: usage of training/pretraining/fine-tuning
>
> Thanks for pointing out this inconsistency, we agree that being consistent with the usage of training, pretraining and fine-tuning will enhance the clarity of our paper. We will carefully review and correct those usages across the paper to make it more consistent.
>
> ## W4,7-9: typos
>
> All fixed in the updated version, thanks!
>
> ## W5: closest work
>
> Both PEQA and QA-LoRA are close to our work in the sense that both are PEFT methods that aim at reducing the memory footprint during training while also maintaining inference efficiency and improved predictive performance compared to PTQ methods.
>
> Note, however, that QA-LoRA constraints matrix A to have the same rows, meaning that it is only designed to be applicable in the case of block-wise weight quantization with a very small group size (e.g., 32). Because of that, it cannot be applied to neither pre-channel nor groupwise weight quantization with bigger groups sizes (64-512), which are common settings for weights.
>
> At the same time, neither PEQA nor our proposed methods have any constraints on the quantization granularity and can equally well be applied to per-channel and even per-tensor weight quantization regimes.
>
> Hence, we consider PEQA to be the closest work and that’s why we decided to reimplement and compare against this baseline.
> We will update the related work section to make this clearer.
>
> ## W6: PEQA trade-offs
>
> Just like LR-QAT, PEQA is also memory- and inference efficient method. However, PEQA has significantly fewer degrees of freedom compared to our method, leading to a subpar predictive performance. At the same time, the accuracy is still consistently better than for PTQ methods.
> Therefore, we see PEQA in Table 1 as follows:
>
> | Method | Accuracy | Memory Efficiency | Inference Efficiency |
> |-|:-:|:-:|:-:|
> | PTQ                | ✕ | ✓ | ✓ |
> | Full-model QAT     | ✓ | ✕ | ✓ |
> | QLoRA              | ✓ | ✓ | ✕ |
> | PEQA               | ✕/✓ | ✓ | ✓ |
> | **LR-QAT (ours)**  | **✓** | **✓** | **✓** |

---

> ### Comment · Reviewer_ipQw · 2024-11-27
> **Raising my score**
>
> Thank you authors for taking into account the suggested changes and providing clarifications to other raised issues. I look forward to seeing the revisions as part of the final manuscript. I am raising my score from 6 to 8.

---

> > ### Author Response · Authors · 2024-11-28
> >
> > Thank you for your updated evaluation and for raising the score, we sincerely appreciate your recognition.
> >
> > Please, refer to the latest general comment on the updated revision of our manuscript, where we tried to include as many recommendations as possible (& promise to include the rest in the final revision).

---

### Official Review · Reviewer_Qdxf · 2024-11-08

**Soundness:** 3
**Presentation:** 3
**Contribution:** 3
**Rating:** 6
**Confidence:** 4

**Summary:**

This paper introduces LR-QAT, a quantization-aware training framework designed for large language models that optimizes memory efficiency without compromising model performance. It incorporates low-rank quantization-aware reparameterization, downcasting, and checkpointing techniques. The proposed technique reduces memory usage significantly often outperforming PQT techniques while allowing integration with such techniques. Additionally, it also does not incur any additional overhead during inference.

**Strengths:**

The paper is well-written and explained clearly. The approach is well-supported by a thorough experimental section, showcasing promising results that validate its efficiency and robustness across multiple large language models.

**Weaknesses:**

The novelty of the approach is somewhat limited as it incorporates several known techniques rather than introducing entirely new concepts. While the paper provides detailed memory and runtime comparisons with full-model QAT (e.g., LSQ), it does not compare these metrics against other implemented baselines like LSQ, or OmniQuant, which limits the assessment of its relative efficiency.

**Questions:**

See weaknesses

---

> ### Author Response · Authors · 2024-11-20
>
> We thank the reviewer for their thoughtful and useful feedback. Please, find our replies to your concerns regarding the novelty and the runtime comparison in the general response to all reviewers.

---

### Author Response · Authors · 2024-11-20
**General response to all reviwers**

We thank all reviewers for their thoughtful and valuable feedback!

We are encouraged that they found our work is well-written and explained clearly (Qdxf, ipQw), well-supported by a thorough experimental section (Qdxf, ipQw, bMgZ), has sufficient ablation studies and theoretical reasoning for method's design choices and observations (ipQw, bMgZ), and has promising results across wide array of models and settings, while greatly reducing the memory footprint and retaining inference efficiency (Qdxf, ipQw, NFjd, bMgZ).

Some reviewers (Qdxf, NFjd) were concerned with the novelty of our work. While we agree that our method is relatively straightforward from an implementation standpoint, it is very effective, easy to use, and is a novel QAT technique that utilizes a low-rank reparameterization for solving the important problem of memory- and inference-efficient LLM quantization. Therefore, we think that despite its simplicity, it is a valuable contribution to the field.

One common question (Qdxf, bMgZ) is on the runtime and memory comparison of LR-QAT against other baselines besides full-model QAT (LSQ). Below you we included a comparison of runtime and memory against PEQA and OmniQuant applied to LLaMA-2 7B. For OmniQuant, we used the publicly available code (https://github.com/OpenGVLab/OmniQuant). Please note that in our detailed memory comparison in Figure 1, PEQA is equivalent to 'Scales only QAT w/ checkpointing'. We will clarify this in the paper.

| Method | GPU mem. [GB] | Time [h] | WT2 w4pc | WT2 w3pc | Zero-shot w4pc | Zero-shot w3pc |
|-|:-:|:-:|:-:|:-:|:-:|:-:|
| Full-model QAT (LSQ), $10^4$ iters | 62.2 (98.5) | 90.2±0.2 | 5.77±0.02 | 6.14±0.01 | 68.96±0.29 | 67.14±0.13 |
| PEQA, $10^4$ iters | 19.9 | 35.6±0.22 | 5.71 | 6.45 | 69.23 | 65.44 |
| OmniQuant | 11.6 | 1.21 | 5.74 | 6.58 | - | - |
| **LR-QAT (ours)**, $10^4$ iters | 20.5 | 42.3±0.13 | 5.66±0.00 | 6.13±0.02 | 69.72±0.32 | 67.70±0.25 |
| **LR-QAT (ours)**, $10^3$ iters | 20.5 | 4.23 | 5.68 | 6.22 | 69.43 | 66.93 |

This is an extended version of Table 3 in the paper. We report the maximum GPU memory with (without) gradient checkpointing and the training runtime on a Nvidia A100 80GB GPU, followed by WikiText-2 (WT2) perplexity (lower is better) and zero-shot accuracy (higher is better) for W4 and W3 per-channel quantization. ± denotes mean and standard deviation over 5 runs with different seeds, where available.

While it is expected that a PTQ technique such as OmniQuant runs faster and consumes less memory than QAT-based techniques, LR-QAT yields better perplexity even with just $10^3$ iterations and is not far from results using $10^4$ iterations (as reported in the paper). We believe that for many use-cases the slightly increased run-time (~1h vs ~4h) might be a desirable trade-off to obtain best accuracy/perplexity. We will include and reference these results in the paper.

We also thank several reviewers (ipQw, NFjd, bMgZ) for pointing out typos and some other minor formatting errors or inconsistencies. We are grateful for their detailed review and will adjust all of them in the next revision.

---

### Author Response · Authors · 2024-11-28
**Updated manuscript**

We have updated our manuscript  to include recommendations from the reviewers.
All the major changes are highlighted in `teal` text for ease of reference.

### **Updates of experimental results during Rebuttal**
* Added and referenced an extended version of Table 3 in Appendix F (Table F2) that includes runtime, memory and accuracy comparison between LR-QAT, Full-model QAT (LSQ), PEQA, and OmniQuant (**Qdxf - W1**, **bMgZ - W3**).
* Generated and included additional OmniQuant baseline results and updated Table 4, updated results sources in Appendix B, and included detailed zero-shot results in Tables C1-C5 (**bMgZ - W2**).
* Included and referenced the RTN initialization ablation study (Table D2) in which we compare min-max vs the best range setting that was used in the main set of results (**bMgZ - W1**).

### **Updates of discussions during Rebuttal**
* Added references of Figure 1 in Sections 3, 4 and a more in-depth discussion on the memory breakdown in Section 5.3 (**ipQw - W1**).
* Corrected the usage of fine-tuning/training/pre-training throughout the paper to be more consistent (**ipQw - W3**).
* Updated Sections 2, 3 to clarify the comparison between PEQA & QA-LoRA and why we considered PEQA to be the closest work to our method (**ipQw - W5**).
* We also updated all typos and minor inconsistencies pointed out by the reviewers (not highlighted for clarity).

### **Further changes for final revision**
As promised, we will incorporate the remaining recommendations from the reviewers in the camera-ready version, namely:
* Include a comparison table that lists what parts of the model are being quantized together with the rest of quantization settings and assumptions across reported methods (**ipQw - W2**).
* Include results of LR-QAT fine-tuned on Alpaca/FLANv2 and comparison vs. QLoRA and QA-LoRA (**NFjd - W2**).
* Extend scale initialization ablation study to also include OmniQuant as an initialization to LR-QAT (**bMgZ - W1**).

We sincerely thank all reviewers for their thoughtful and valuable feedback, which has been instrumental in enhancing the quality of our manuscript. We look forward to the reviewers' favorable consideration and remain grateful for their valuable feedback.

---

### Meta-Review · Area_Chair_tDSp · 2024-12-26

**Metareview:**

This paper proposes a combination of low-rank and quantization-aware training for LLMs, which  incorporates low-rank quantization-aware reparameterization, downcasting, and checkpointing techniques. The strength of the paper is the good empirical results, showing the proposed framework is better than previous quantization/low-rank training in several settings, while there are two major weaknesses pointed out by the reviewers: (1) Since the proposed method is based on a combination of several existing techniques, the novelty of the paper is limited, and (2) Several reviewers pointed out some missing numbers in the results and missing baselines/datasets, which makes it harder to conclude whether the proposed method is generally better than previous ones in general.

Since the novelty is limited and the current empirical results are not that complete, we recommend to reject the paper and encourage the authors to improve the experimental results and resubmit their paper to another top ML conference.

**Additional Comments On Reviewer Discussion:**

Reviewers raised several questions about both novelty and empirical results in their reviews. Unfortunately, due to the difficulty of running more experiments in a short rebuttal period, two of the reviewers still have concerns about the empirical comparisons after author-reviewer discussions.

---

### Decision · Program_Chairs · 2025-01-22

Reject